

# Progress in the quantification of aerosol-cloud interactions estimated from the CALIPSO-CloudSat-Aqua/MODIS record

Zhujun Li[1,2], David Painemal[2], Yan Feng[3], and Xiaojian Zheng[3]

[1]Analytical Mechanics Associates, Inc., Hampton, 23666, USA
[2]NASA Langley Research Center, Hampton, 23666, USA
[3]Argonne National Laboratory, Lemont, 60439, USA

*Correspondence to:* David Painemal (david.painemal@nasa.gov)

**Abstract.** Aerosol-cloud-precipitation interactions are assessed over the non-polar ocean from more than 11 years of combined Aqua-MODIS, CALIPSO-CALIOP, and CloudSat products. The analysis first shows the benefit of incorporating vertically resolved aerosol extinction coefficient ($\sigma_{ext}$) in aerosol-cloud interactions (ACI) assessments, demonstrating that: $\sigma_{ext}$ vertically collocated with the cloud layer correlates best with cloud droplet number concentration ($N_d$), column-integrated aerosol optical depth (AOD) cannot explain the $N_d$ variability in the extratropics, and the S-shape of the AOD-$N_d$ relationship reported in previous studies is an unphysical feature that arises from using AOD as aerosol proxy over oceanic regions. ACI metric, estimated as the log-scale regression between $\sigma_{ext}$ vertically collocated with the cloud layer ($\sigma_{ext}^{CL}$) and MODIS $N_d$ reveals that the eastern Pacific is the region with the strongest ACI, followed by the Southern Ocean. The susceptibility of clouds to changes in their liquid water path (LWP) and frequency of precipitation followed a 2-step calculation by combining the $N_d$ - $\sigma_{ext}^{CL}$ regression (ACI) with the regression between these macrophysical variables and $N_d$. LWP susceptibility is negative (LWP decreases with aerosol loading), and statistically significant over the eastern Pacific, eastern Atlantic, and extratropics. In contrast, vast areas of the tropical and subtropical ocean feature negligible changes in LWP with aerosol. Precipitation frequency susceptibility is negative, but the values are only significant over the coastal eastern Pacific and Atlantic. The findings suggest that previous modeling assessments relying on AOD may need to be revisited by taking advantage of the synergy between passive and active sensors.

## 1 Introduction

Estimates of aerosol-cloud interactions (ACI) and cloud adjustments are critical for understanding the role of aerosols and clouds in climate and for testing the ability of models to simulate these susceptibilities. During the past decades, numerous studies have taken advantage of multi-year satellite observations for investigating ACI and cloud rapid adjustments in liquid boundary layer clouds (e.g. Myhre et al., 2007; Quaas et al., 2009; Chen et al., 2014). Although questions remain about the appropriateness of evaluating changes in radiative forcing since pre-industrial times with the use of the current satellite data record (e.g Mülmenstädt et al., 2024), a more fundamental question to be addressed is whether linear regressions between satellite-derived cloud and aerosol properties capture meaningful physical mechanisms. An encouraging line of evidence is the positive linear correlation observed between satellite aerosol optical depth (AOD) and cloud droplet number concentration (e.g. Quaas et al., 2008), which is generally consistent with airborne observations (e.g. Sorooshian et al., 2019), and in agreement with expectations of the first aerosol indirect effect (increase in cloud droplet concentration with aerosol concentration). This correlation consistency appears to be, in part, attributed to the good skill of satellite retrievals to replicate features observed by ground-based and in-situ platforms, especially over the ocean (e.g. Levy et al., 2013; Painemal et al., 2019, Gryspeerdt et al., 2022). However, fortuitous non-causal aerosol-cloud correlations could impact the interpretation of satellite-based statistics and the way they are used for understanding real



physical processes. Complexities arise in particular from the use of column-integrated AOD, as its adequacy for representing aerosol concentration or cloud condensation nuclei (CCN) in aerosol activation to cloud droplets have been called into question (Shinozuka et al, 2015; Stier, 2016). This is because AOD (or aerosol index) does not uniquely represent aerosol concentration or CCN concentration, as variations in aerosol composition, particle size distribution, and optical properties can yield the same AOD for different aerosol concentrations. A second limitation is the inability to disentangle the contributions of different aerosol layers to the total AOD, which prevents any meaningful vertical collocation between aerosol and cloud layer. These limitations are likely responsible for notable differences between in-situ- and satellite-based aerosol-cloud relationships. For example, the observed logarithmic AOD–$N_d$ relationship from satellites resembles a S-curve: $N_d$ features modest variations with AOD for small AOD values, followed by a rapid linear increase of $N_d$ with AOD, and culminating in $N_d$ values that remain nearly constant for high values of AOD (Gryspeerdt et al., 2016). While it is generally assumed that the insensitivity of $N_d$ to high AOD is likely the result of less aerosol activation in highly polluted environments with substantial CCN availability, the weak $N_d$–AOD dependency for pristine environments is difficult to interpret without invoking large uncertainties in AOD for regions with small aerosol burden. The weak relationship between $N_d$ and AOD for pristine areas is particularly troubling especially considering the widespread occurrence of regions with low AOD over the ocean, which is precisely where one should expect a substantial occurrence of boundary layer clouds. These results are, again, at odds with multiple field campaigns, which consistently identify linear changes of aerosol concentration with $N_d$ for a wide range of aerosol concentrations (e.g. McFarquhar et al., 2022; Painemal and Zuidema 2013; Gupta et al., 2022; Zheng et al., 2024).

In addition to limitations in the physical information derived from satellite observations, retrieval artifacts can also impact the interpretation of aerosol-cloud linear regressions. For instance, recent studies have warned about biases of aggregating satellite observations without removing pixels more prone to uncertainties (Painemal et al., 2025). Moreover, enhanced aerosol swelling, cloud contamination, and three-dimensional radiative effects can affect the collocated satellite AOD pixels near cloud edges (Varnai and Marshak, 2009). To advance in the ACI quantification, Painemal et al. (2020) propose the use of vertically resolved satellite aerosol retrievals, with the objective of isolating the aerosol layer closer in altitude to the cloud layer from the rest of the aerosol column. More specifically, the incorporation of Cloud-Aerosol Lidar and Infrared Pathfinder Satellite Observations (CALIPSO) based aerosol retrievals to the analysis is advantageous for minimizing sensitivities to 3D radiative effects and cloud contamination. Regrettably, the application of spaceborne lidar observations to the ACI computation is still surprisingly lacking. Motivated by the proof-of-concept introduced in Painemal et al. (2020) we take advantage of more than 11 years of collocated daytime CALIPSO aerosol properties, MODerate resolution Imaging Spectroradiometer (MODIS) cloud retrievals, and CloudSat precipitation estimates to quantify ACI over the non-polar ocean. Our overarching objectives are: a) to investigate the benefits and shortcomings of using vertically resolved aerosol properties, and b) to compute metrics of cloud susceptibilities of ACI and cloud susceptibility over the non-polar oceans.

## 2 Data and methods

### 2.1 Satellite products

The dataset for this study comprises daytime observations from Cloud-Aerosol LIdar with Orthogonal Polarization (CALIOP) on the CALIPSO, the CloudSat's Cloud Profiling Radar (CPR), and the MODIS on Aqua, from July 2006 to December 2017, for most of the period for which the 3 satellites flew in formation as a part of the A-Train constellation.





### 2.1.1 CALIOP

Aerosol retrievals are taken from a research product described in Painemal et al. (2019) that combines CALIPSO attenuated backscattering coefficient with an AOD product derived from the CALIOP's ocean surface return based on the Synergized Optical

Depth of Aerosols algorithm (SODA, Josset et al., 2008), described in Painemal et al. (2019). This choice of CALIPSO-based dataset responds to limitations of the standard CALIPSO product associated with the requirement of detecting aerosol layers and categorizing them into a limited number of aerosol types, adversely affecting the availability of CALIPSO AOD and extinction coefficient datapoints, and potentially biasing the retrievals especially when aerosol type misclassification occurs (e.g. Kim et al., 2017). The derivation of aerosol extinction coefficient ($\sigma_{ext}$) profiles at 60 m vertical resolution makes use of the attenuated

backscattering coefficient and SODA AOD to invert the lidar equation by applying the Fernald-Klett iterative algorithm (Fernald, 1984). The method is described in detail in Painemal et al. (2019) and Painemal et al. (2020), and applications for evaluating CALIPSO operational aerosol products are reported in Li et al. (2022). The CALIPSO-based aerosol products are spatially averaged to the standard 5-km resolution of CALIPSO and integrated into the analysis. To simplify the notation, we refer to the CALIPSO SODA aerosol retrievals as CALIOP-S. To reduce the effect of cloud contamination and signal enhancement due to

aerosol swelling near cloud edges, we remove from the analysis 5-km spatial averages with CALIOP-S cloud fraction greater than zero.

Cloud top height from CALIPSO version 4.2 (LID_L2_01kmCLay-Standard product) at 1-km resolution is added to the analysis, as it provides accurate detection of cloud top height (ZT). Because the focus of our study is boundary layer clouds (low clouds), we select pixels with ZT < 3km, and compute 5-km spatial averaging, with the corresponding 5-km cloud fraction calculated as

the fraction of 1-km low-cloud pixels within the 5-km scanline section.

### 2.1.2 Aqua MODIS

MODIS cloud properties correspond to pixel-level cloud retrievals obtained from the Cloud and Earth's Radiant Energy System (CERES) Edition 4 product (Minnis et al., 2020). Variables ingested into the analysis include cloud droplet effective radius ($r_e$), optical depth ($\tau$), temperature, and height (pressure). Liquid water path (LWP) is estimated using the relationship $LWP = \frac{5}{9} \cdot \rho \cdot$

$r_e \cdot \tau$, with ρ denoting the liquid water density. To limit the analysis to low clouds, we only aggregate liquid-phase MODIS pixels with cloud tops below 3 km. $N_d$ is calculated at 1-km (pixel-level) resolution, using the adiabatic formulation described in Painemal (2018) and Grosvenor et al. (2015):

$$N_d = \Gamma^{1/2} \frac{10^{1/2}}{4\pi \rho_w^{1/2} k} \frac{\tau^{1/2}}{r_e^{5/2}} \tag{1}$$

The parameter Γ in (1) is the adiabatic condensation rate of water vapor with height (Albrecht et al., 1990), which is a function of

105 temperature and pressure. Departures from the adiabatic Γ value are not considered here due to a lack of understanding of how to estimate this adjustment with satellite data. The parameter k is the ratio between the volume radius and $r_e$ and is assumed constant at 0.8 (Martin et al., 1994).

### 2.1.3 CloudSat Cloud Profiling Radar

CloudSat parameters are obtained from the 2B-GEOPROF Release 05 product. Cloud reflectivity is utilized in our analysis for precipitation detection. To minimize the effect of artifacts and surface echo, we used the CloudSat cloud mask for retaining samples with good and strong echoes (mask value of 30 or 40). The CloudSat maximum radar reflectivity of the cloud column with tops below 3 km ($Z_{max}$) is used to categorize the low-cloud precipitation rate for $Z_{max}$ > -15 dBZ. The impact of additional precipitation categorization (drizzle: -15< $Z_{max}$<=-7), light rain: -7< $Z_{max}$<=0, and rain: $Z_{max}$>0) is discussed in Section 4.




## 2.2 Data matching and additional averaging

The data matching methodology follows Painemal et al. (2020) and is designed to combine datasets with different spatial resolutions, as well as to reduce potential sources of uncertainties that could otherwise impact our analysis. Briefly, the matching is conducted for individual 25-km segments along the CALIPSO ground track (Fig. 1), with the goal of creating a dataset of MODIS, CALIOP, and CloudSat retrieval aggregated to a 25-km resolution. We start by averaging the CALIPSO cloud height to yield a single value per 25-km segment, with values retained for averages constructed with at least 20 % of cloudy observations for the 25-km scanning line to guarantee a significant number of samples in the computation of cloud top height. Next, cloud-free CALIOP-S aerosol extinction coefficients at 5-km resolution are spatially averaged over the same 25-km segment. Lastly, the closest CloudSat CPR pixels to the 25-km line are combined to derive a probability of precipitation (POP) defined as the fraction of precipitating pixels of the total cloudy pixels within the segment, with precipitation defined for samples with $Z_{max}$ greater than -15 dBZ. Note that CloudSat and CALIPSO ground-tracks in Figure 1 are not identical (e.g., Mace and Zhang, 2015), yet the discrepancy is much less than the 10-km cross-track distance in Fig. 1.

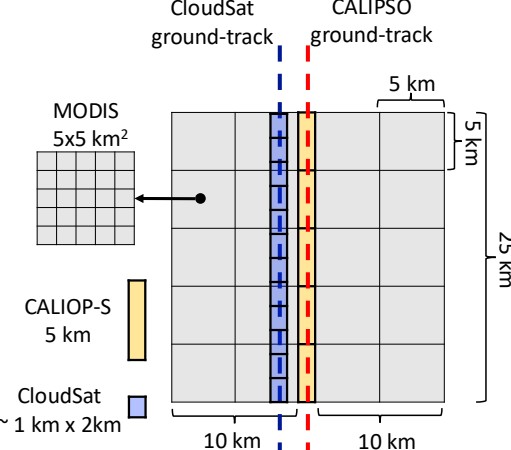

**Figure 1: Spatial collocation of the 3 datasets along a 25-km CALIPSO along-track segment. CloudSat footprints are being represented without oversampling. CALIPSO CALIOP-S cross-track footprint is less than 100 m.**

Considering that cloud retrievals and $N_d$ come from MODIS, we take a number of steps to reduce retrieval biases. We first match MODIS pixels with the 5-km CALIPSO pixel (Fig. 1 yellow block) by using 5-pixel x 5-pixel MODIS boxes, with 2 boxes east and 2 boxes west of the CALIPSO ground-track (Figure 1, gray squares). Second, for each of these MODIS data boxes, the 5-km² low-cloud fraction is calculated as the number of liquid phase cloudy points with cloud top heights of less than 3 km divided by the total number of points. Third, the 20 5x5 MODIS boxes are individually averaged and averaged boxes with cloud fraction greater or equal than 80% are retained for future averaging. Then, the averaged MODIS boxes centered along the 25-km CALIPSO track segment are finally averaged to produce a single cloud value collocated at the 25-km CALIPSO along-track resolution. At this resolution, averaged MODIS data are used in the analysis when the solar zenith angle is less than 65° and the mean cloud optical depth is greater than 2.0, which helps reduce uncertainties in optically thin clouds (Painemal et al., 2025). Lastly, we only analyzed samples with CALIOP-S AOD greater than 0.05 to reduce uncertainties in the derivation of very low AOD (Painemal et al., 2019).



A final threshold applied to the 25-km aggregated data corresponds to limiting the analysis to MODIS grids with low-cloud fraction
equal to or less than 90%. This upper limit enables the removal of 25-km grids with aerosols fully embedded in cloudy regions,
which are more severely affected by aerosol swelling in areas with peaks in humidity (Painemal et al., 2020).

### 2.3 Aerosol layers

For evaluating the impact of aerosol layers in the ACI quantification, we compute from the 25-km aerosol extinction coefficient
horizontal averages, the vertically averaged $\sigma_{ext}$ for three 300-m atmospheric layers (Painemal et al., 2020): near-surface (SFC),
cloud-level (CL), and free troposphere (FT). Near-surface $\sigma_{ext}$ ($\sigma_{ext}^{SFC}$) is estimated as the vertical average value between the height
43 m and 343 m above the sea level. Cloud-level average $\sigma_{ext}$ ($\sigma_{ext}^{CL}$) is computed as the average for the 300-m layer between 360
m and 60 m below the mean cloud top height (25-km CALIOP ZT). Free tropospheric $\sigma_{ext}$ ($\sigma_{ext}^{FT}$) is the 300-m layer average
between the altitude 60 m and 360 m above the mean CALIPSO cloud top height. The 60 m departure from ZT for the $\sigma_{ext}^{CL}$ and
$\sigma_{ext}^{FT}$ calculation is intended to minimize the influence of uncertainty in ZT retrievals by limiting the contribution of samples in the
free troposphere and boundary layer to the $\sigma_{ext}^{CL}$ and $\sigma_{ext}^{FT}$ averages, respectively.

### 2.4 Satellite susceptibilities

The aerosol-cloud interactions (ACI) metric is defined as the fractional change of $N_d$ in response to the fractional change of aerosols
(Eq. 2). In this study, the aerosol component is represented by the layer-averaged $\sigma_{ext}$, and ACI expressed as:

$$ACI = \frac{d\ln(N_d)}{d\ln(\sigma_{ext})}, \tag{2}$$

The computation of cloud adjustments (susceptibilities) to aerosols acknowledges the fact that cloud properties (liquid water path,
precipitation, and cloud fraction) are modulated by $N_d$, which is, in turn, sensitive to variations in aerosol properties ($\sigma_{ext}$). It follows
that LWP, precipitation (POP) and cloud fraction (CF) susceptibilities to aerosol –$S_{LWP}$, $S_{POP}$ and $S_{CF}$ respectively– can be
expressed as:

$$S_{LWP} = \frac{d\ln(LWP)}{d\ln(\sigma_{ext})} = \frac{\partial\ln(LWP)}{\partial\ln(N_d)} \cdot \frac{\partial\ln(N_d)}{\partial\ln(\sigma_{ext})}, \tag{3a}$$

$$S_{POP} = \frac{d\ln(POP)}{d\ln(\sigma_{ext})} = \frac{\partial\ln(POP)}{\partial\ln(N_d)} \cdot \frac{\partial\ln(N_d)}{\partial\ln(\sigma_{ext})}, \tag{3b}$$

$$S_{CF} = \frac{d\ln(CF)}{d\ln(\sigma_{ext})} = \frac{\partial\ln(CF)}{\partial\ln(N_d)} \cdot \frac{\partial\ln(N_d)}{\partial\ln(\sigma_{ext})}, \tag{3c}$$

Defining cloud susceptibilities due to cloud microphysical changes as: $S_{LWP}^{Nd} = \frac{\partial\ln(LWP)}{\partial\ln(N_d)}$, $S_{POP}^{Nd} = \frac{\partial\ln(POP)}{\partial\ln(N_d)}$, and $S_{CF}^{Nd} = \frac{\partial\ln(CF)}{\partial\ln(N_d)}$, we
can simplify the notation and express the overall susceptibility due to aerosols as:

$$S_{LWP} = S_{LWP}^{Nd} \cdot ACI, \tag{4a}$$

$$S_{POP} = S_{POP}^{Nd} \cdot ACI, \tag{4b}$$

$$S_{CF} = S_{CF}^{Nd} \cdot ACI, \tag{4c}$$

ACI, $S_{LWP}^{Nd}$, $S_{POP}^{Nd}$, and $S_{CF}^{Nd}$ are calculated as the linear regression between the natural logarithm of cloud and aerosol properties,
following a binning method described in the following sections.



## 3 Results

### 3.1 Impact of the aerosol layer selection

The three different values of layer aerosol extinction coefficient ($\sigma_{ext}^{SFC}$, $\sigma_{ext}^{CL}$, and $\sigma_{ext}^{FT}$) are utilized to determine whether a specific aerosol layer covaries the strongest with $N_d$. To this end, we compute maps of linear correlation coefficient (r) for the matched observations using a 5˚x5˚ regular grid. First, we group the control variable ($\sigma_{ext}$) in 20 quantiles, with the goal of determining $\sigma_{ext}$ bin sizes common to all 5˚x5˚ regions. Next, we average $N_d$ as a function of the 20 $\sigma_{ext}$ bins. To eliminate spurious results associated with reduced sampling in each 5˚x5˚ grid, we only use binned $\sigma_{ext}$ - $N_d$ when they are created with at least five paired samples per bin, and the total number of valid bins is at least 10, totalling at least 50 datapoints for regression calculation. The maps in Fig. 2 depict r for $N_d$–$\sigma_{ext}^{SFC}$ (Fig. 2a), $N_d$–$\sigma_{ext}^{CL}$ (Fig. 2b), and $N_d$–$\sigma_{ext}^{FT}$ (Fig. 2c), with gray areas representing grids with insufficient number of samples or valid bins to perform the calculation. Overall, the analysis shows that $N_d$ correlates the highest with cloud-level $\sigma_{ext}$ with overall correlation coefficient greater than 0.7 over vast oceanic regions, except for the tropical Pacific, where the correlations are modest (Fig.2b). Surface-layer $\sigma_{ext}$ is positively correlated to $N_d$ in the littoral regions of South Atlantic, the western Africa, Indian Ocean, and western North Pacific, however, the correlations are weaker than those for $N_d$-$\sigma_{ext}^{CL}$. The $N_d$ is least correlated to the free tropospheric $\sigma_{ext}$ (Fig. 2c), but with a few patches of r > 0.5 over the east coast of South America and southern Africa. Overall, the findings in Fig. 2 support the hypothesis formulated in Painemal et al. (2020) and Stier (2016) in that isolating the aerosol layer closer to the cloud deck is central for a more rigorous assessment of aerosol-cloud interactions. For the rest of this study, we will primarily center our attention on the relationship between $\sigma_{ext}^{CL}$ and other cloud quantities.



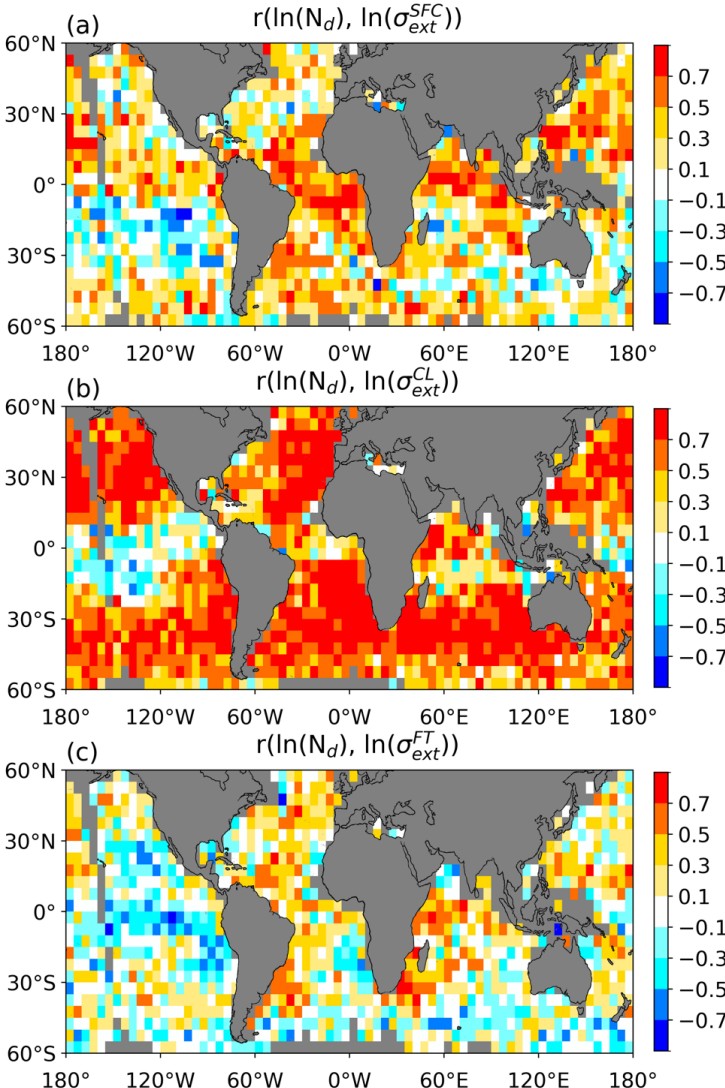

**Figure 2: Gridded maps of correlation coefficient between MODIS $N_d$ and (a) surface layer $\sigma_{ext}$ ($\sigma_{ext}^{SFC}$), (b) cloud level $\sigma_{ext}$ ($\sigma_{ext}^{CL}$), and**
**(c) free tropospheric $\sigma_{ext}$ ($\sigma_{ext}^{FT}$). Correlations are estimated after applying the natural logarithm to the variables.**

Having demonstrated that the aerosol extinction coefficient adjacent to the cloud-layer altitude is the parameter that best correlates

with $N_d$, we assess the benefits of applying $\sigma_{ext}^{CL}$ to the analysis relative to the use of standard AOD as a control variable. For this

purpose, we consider the relationship between CALIOP-S aerosol retrievals and $N_d$ for five latitudinal bands and compare the

aerosol–$N_d$ relationship for the 2 control variables: $\sigma_{ext}^{CL}$ and AOD. For each regional band, we average $N_d$ as a function of 50

CALIOP-S aerosol bins with equal number of data points. It is noteworthy to mention that because $\sigma_{ext}$ in the CALIOP-S data

product is constrained using AOD (Painemal et al., 2020), different results can only be attributed to the use of vertically resolved

versus vertically integrated quantities, rather than product and algorithm discrepancies. A key characteristic depicted in Fig. 3a is

that the shape of the $N_d$ -$\sigma_{ext}^{CL}$ relationship can be generally captured by a linear fit, with some departures for the 10% smallest

aerosol extinction coefficients (<0.01 km⁻¹), which are values within the retrieval uncertainty range (Painemal et al., 2019).





Moreover, the strong relationship is observed across all the latitudinal bands, with Spearman correlation coefficients greater than 0.98. On the other hand, $N_d$ shows little sensitivity to AOD for specific AOD regions (Fig. 3b). For example, the 40°-60° latitude bands exhibit modest changes in $N_d$ with AOD for AOD < 0.2 (Fig. 3b, magenta and black), with overall Spearman correlation coefficient of 0.37 (40°N-60°N) and 0.16 (40°S-60°S). For other bands (20°N-40°N and 20°S-20°N), the sensitivity of $N_d$ is modest for AOD>0.2, with variations of less than 10 cm$^{-3}$ in the tropical region (Fig., 3b, blue triangles). In sum, the analysis reveals that either, AOD is poorly correlated to $N_d$ or the $N_d$-AOD relationship is poorly represented by a linear fit. Moreover, because the flattening of the $N_d$ curve with AOD is not observed in $\sigma_{ext}$, this suggests that the S-shape curve between $N_d$ and AOD reported in a number of studies may not be the manifestation of microphysical processes, rather it reflects the inadequacy of AOD as an aerosol proxy for ACI studies, especially for higher $N_d$.

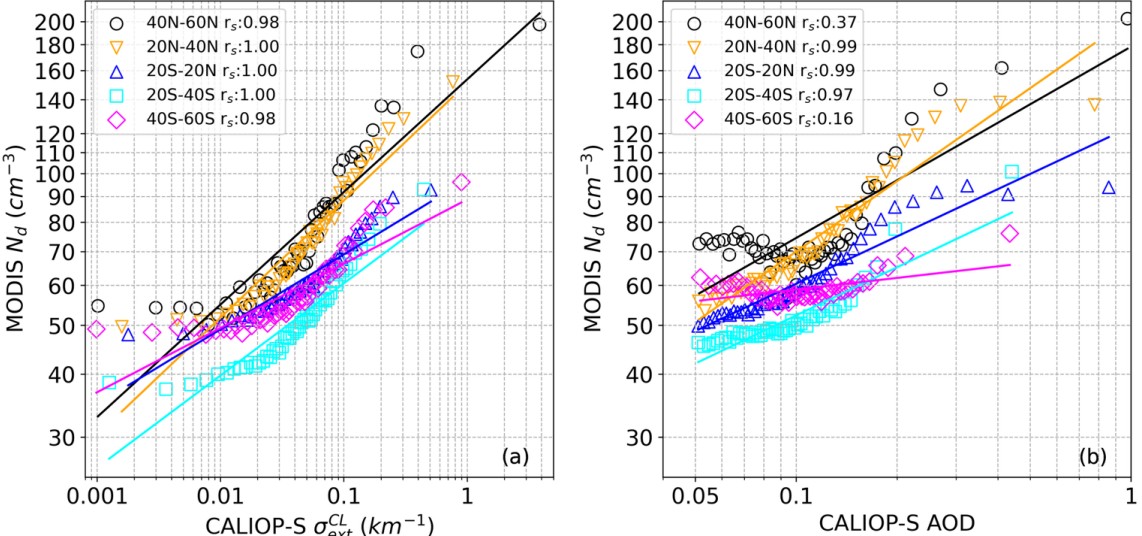

**Figure 3:** MODIS $N_d$ as a function of (a) CALIOP-S $\sigma_{ext}^{CL}$ and (b) AOD. The relationships are shown for five latitude bands: 40°N to 60°N (black circle); 20°N to 40°N (gold triangle); 20S to 20°N (blue triangle); 20°S to 40°S (cyan square); 40°S to 60°S (magenta diamond). The linear best fit of each latitude band, represented by the line of corresponding color is estimated from the variables in logarithmic scale.

### 3.2 Aerosol-cloud interactions

Given the benefits of using $\sigma_{ext}^{CL}$ for studying the impact of aerosols on $N_d$, we apply Eq. 2 to estimate ACI via the linear regression between $N_d$ and $\sigma_{ext}^{CL}$. Fig. 4 shows the ACI map, computed following the same 20-bin methodology and spatial resolution applied to the construction of the 5°x5° correlation maps in Fig. 2. Statistically significant values are positive over most of the non-polar oceans, consistent with the notion that more aerosols drive an increase in cloud droplet number concentrations. The highest fractional changes of $N_d$ with $\sigma_{ext}^{CL}$ are found in the coastal southeast Pacific, the northeast, and southeast Atlantic. These ACI peaks coincide with the location of subtropical stratocumulus cloud regimes, which have shown the largest sensitivity to changes in their shortwave fluxes due to perturbations in $N_d$ (Painemal 2018; Zhang and Feingold, 2023). Other regions with high ACI include the northeast Pacific Ocean and the Southern Ocean within the 60°W-140°E zonal band. In contrast, values statistically indistinguishable from zero are found over vast regions in the tropical ocean, where shallow cumulus clouds more frequently occur.

It is also noteworthy that regions with high ACI in the southeast Atlantic and eastern Pacific, are also associated with modest precipitation occurrence (Fig. 5). Indeed, global ACI for non-precipitating ($Z_{max}$ < -15 dBZ) and precipitating ($Z_{max}$ > -15dBZ)



segments is 0.13 and 0.08, respectively. However, the spatial variability of the ACI map does not exactly covaries with POP depicted in Fig. 5, especially in the Southern Ocean, which features both high ACI and precipitation occurrence.

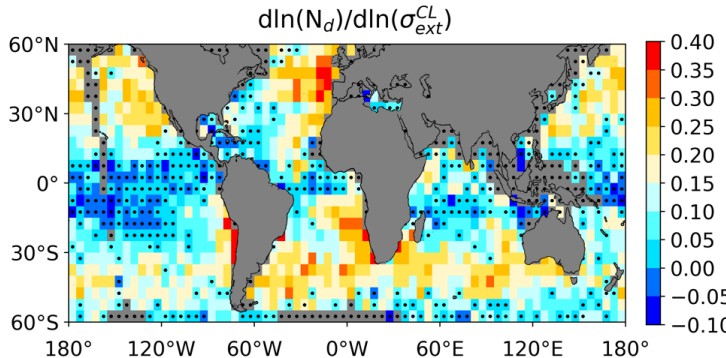

**Figure 4: Gridded map of ACI index (d ln ($N_d$)/ d ln ($\sigma_{ext}^{CL}$) ). Black dots indicate grids that are statistically indistinguishable from zero, according to a Student's t test at 95% confidence level.**

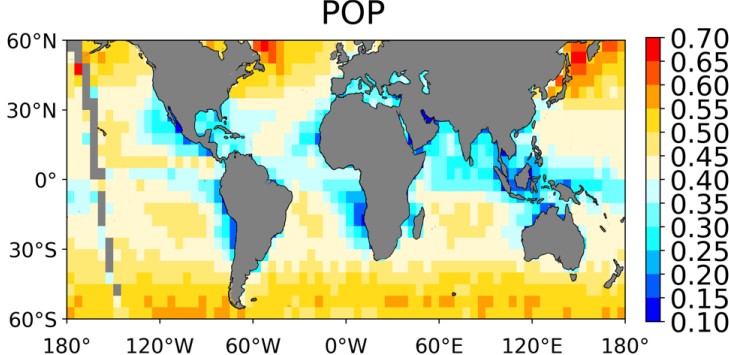

**Figure 5: Mean frequency of precipitation occurrence for cloudy observations from CloudSat, with precipitating samples defined as having $Z_{max} > -15$ dBZ.**

### 3.3 LWP susceptibility

The estimation of susceptibilities follows the regression method used for ACI. We start with the LWP-$N_d$ sensitivity term of Eq. 4a ($S_{LWP}^{Nd} = \frac{\partial \ln(LWP)}{\partial \ln(N_d)}$), as depicted in Fig. 6a. The map reveals two regimes over the ocean: 1) LWP increases with $N_d$ over the tropics, and 2) LWP decreases in the subtropics and extratropics, with minima in high latitudes. This pattern is consistent with the results in Gryspeerdt et al. (2019), although the negative/positive $S_{LWP}^{Nd}$ contrast is more striking in our analysis, likely due to the less stringent data filtering applied in our study, which favors a wider dynamic range in LWP than that in Gryspeerdt et al. (2019).

A closer look at four specific 20°x20° regions with positive and negative signs of $S_{LWP}^{Nd}$, uncovers how the LWP dependency on $N_d$ varies for different ranges of LWP (Fig. 7). For the tropical areas (Fig. 7, black circles and magenta triangles), the strong positive LWP-$N_d$ correlation is observed for low values of LWP with $N_d$. Conversely, the negative correlation in other regions is characterized by larger LWP for low $N_d$, decreasing to LWP < 40 g m$^{-2}$ for $N_d$ > 100 cm$^{-3}$. It is interesting to note that the inverted-V shape of the LWP-$N_d$ relationship reported from global statistics, is less pronounced or even absent at regional scales in Fig. 7.



It is, thus, the strong inverted-V shape derived in previous studies (e.g. Gryspeerdt et al., 2019) the likely result of combining regions with negative and positive $S_{LWP}^{Nd}$ into the same analysis. This explanation is similar to that in Arola et al. (2022), which postulates that natural heterogeneity can contribute to the misinterpretation of the LWP–$N_d$ relationship.

LWP susceptibility $S_{LWP}$, is finally estimated as the product between $S_{LWP}^{Nd}$ and ACI (Fig. 6b). The $S_{LWP}$ map features an overall negative susceptibility, indicating that the aerosol effect on LWP is a net reduction in LWP with an aerosol increase. It is also

interesting that the susceptibility pattern is mainly driven by extratropical clouds in the Southern Ocean, eastern Pacific and Atlantic oceans. On the other hand, the susceptibility in the tropics and in parts of the subtropical open ocean is negligible.

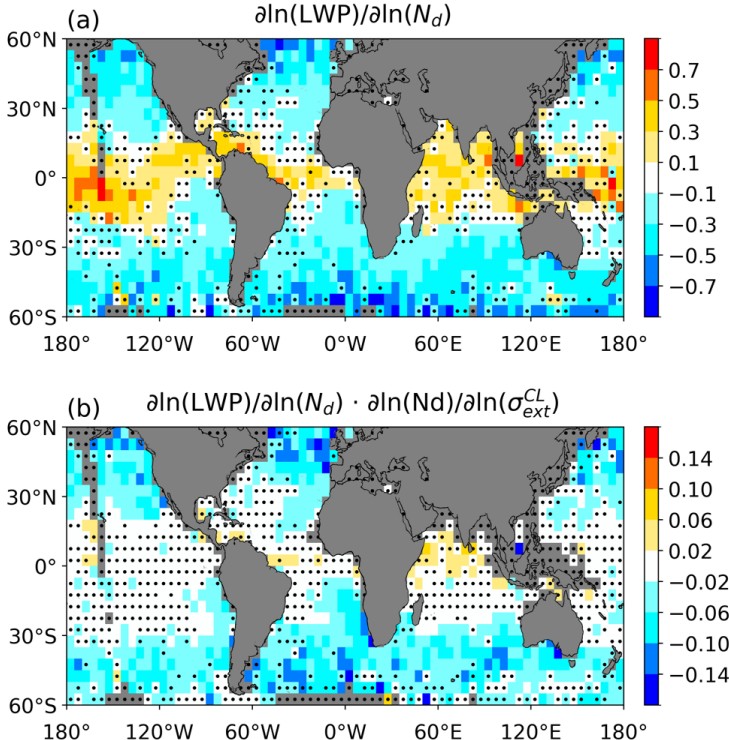

**Figure 6: Gridded maps of (a) susceptibility of LWP to $N_d$ or $S_{LWP}^{Nd} = \frac{\partial \ln(LWP)}{\partial \ln(N_d)}$; and (b) overall LWP susceptibility to aerosols estimated as $S_{LWP} = S_{LWP}^{Nd} \cdot ACI$. Black dots in (a) indicate grids that are statistically indistinguishable from zero, according to a Student's t test at 95% confidence level, whereas dots in (b) represent boxes when at least one metric (ACI or $S_{LWP}^{Nd}$) is statistically indistinguishable from zero.**





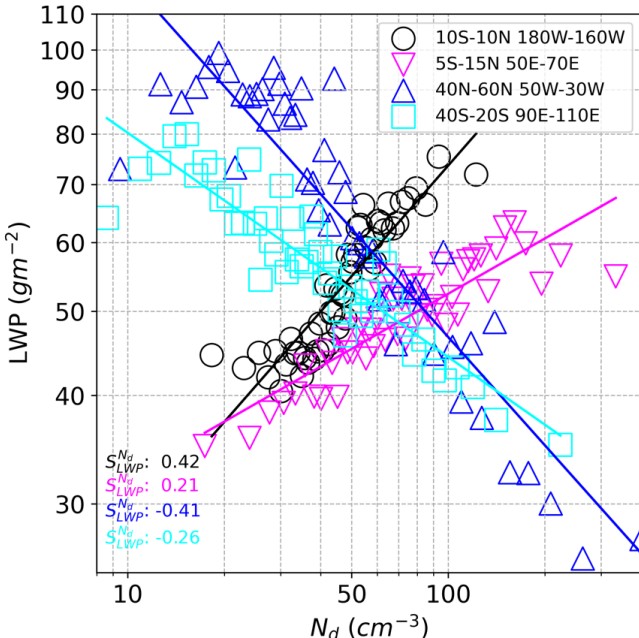

**Figure 7: LWP-$N_d$ relationship for 20°x20° regions with opposite sign slope. Central Pacific: 10°S-10°N, 180°W-160°W (black circles); tropical Indian Ocean: 5°S-15°N, 50°E-70°E (magenta inverted triangles); north Atlantic: 40°N-60°N, 50°W-30°W (blue triangles); and**
**Southern Ocean: 40°S-20°S, 90°E-110°E (cyan square).**

### 3.4 POP (precipitation) susceptibility

The first step for estimating precipitation (POP) sensitivity is to quantify the POP-$N_d$ sensitivity ($S_{POP}^{Nd}$, Fig. 8a). Due to the lack of precipitating samples and surface clutter in the CloudSat product, it is not possible to consistently estimate $S_{POP}^{Nd}$ for all regions.

Moreover, when the data yield allows for the estimate of $S_{POP}^{Nd}$, the values are insignificant for most oceanic areas (Fig. 8a). For the regions with statistically significant $S_{POP}^{Nd}$, the total POP sensitivity to $N_d$ is mostly negative, with the strongest susceptibilities over the eastern Pacific and southeast Atlantic. These regions are also those with statistically significant values of overall precipitation susceptibility due to aerosols (Fig. 8b). For these stratocumulus cloud regimes, the negative susceptibility is consistent with the notion that aerosols suppress precipitation simulated by numerical models (Mülmenstädt et al, 2024).



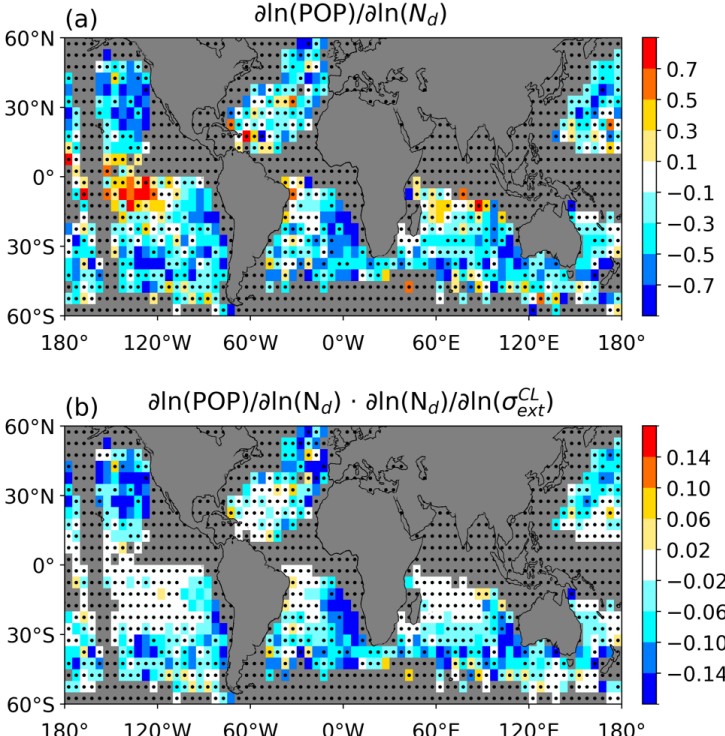

**Figure 8: Gridded maps of (a) susceptibility of POP to $N_d$ or $S_{POP}^{Nd} = \frac{\partial \ln(POP)}{\partial \ln(N_d)}$; and b) overall POP susceptibility to aerosols estimated as $S_{POP} = S_{POP}^{Nd} \cdot ACI$. Black dots in Fig. 8a indicate grids that are statistically indistinguishable from zero, according to a Student's t test at 95% confidence level, whereas dots in Fig. 8b represent boxes when at least one metric (ACI or $S_{POP}^{Nd}$) is statistically indistinguishable from zero.**

## 4 Discussion

Given the novel results presented in this study, it is pertinent to revisit previous ACI assessments based on Aqua MODIS AOD. For this purpose, we take 5 years of daily Collection 6 MODIS Level 3 (L3) Atmosphere Gridded Product (MYD08_D3) and we compute ACI as $ACI_{AOD} = \frac{d \ln(N_d)}{d \ln(AOD)}$, following the binning methodology used for the CALIOP-MODIS ACI calculations. MODIS L3 ACI map (Fig. 9) features a regional distribution that depart from the MODIS-CALIOP (Fig. 4). An important difference between both ACI estimates is in their magnitude, with MODIS L3 exhibiting values twice as large as those from MODIS-CALIOP. Because the functional relationship between AOD and $N_d$, and AOD and $\sigma_{ext}$ are highly non-linear (Fig. 3b and Painemal et al., 2020), with the disadvantages of using AOD previously discussed, comparing ACI$_{AOD}$ magnitudes does not provide meaningful information. Instead, our focus is on the interpretation of regional changes relative to the global map. In this regard, a key difference is the negligible ACI$_{AOD}$ in the extratropics, especially in the Southern Ocean and north of 40˚N. This contrasts with the local maximum observed over the same region for the $\sigma_{ext}^{CL}$-based ACI (Fig. 4). Moreover, except for the region west of Australia, the stratocumulus subtropical regions of the eastern Pacific and Atlantic show modest ACI in the AOD-based calculation, in disagreement with the analysis of Figure 4 and with in-situ observations (e.g. Kang et al., 2021; Gupta et al., 2022; Sorooshian et al., 2019; Zheng et al., 2024). Fig. 9 raises the concern, once again, that AOD-based analyses might be misrepresenting ACI



and, thus, the use of AOD for evaluating models could misguide modelers about the physical processes that need to be refined, or retained, in models.

An aspect not explored in this study is the relationship between aerosol extinction coefficient and aerosol concentration and how it would impact the ACI calculations. While empirical relationships do show a close linear log-scale relationship between boundary

layer aerosol concentration and aerosol extinction coefficient (e.g. Shinozuka, et al., 2015) significant regional variations are expected to affect the extinction-to-CCN conversion. An additional difficulty is accounting for the effect of ambient relative humidity in controlling the aerosol hygroscopicity and optical properties (Gasso et al., 2000), which is also dependent on aerosol size and chemical composition. To circumvent the issue of applying empirical equations for correcting $\sigma_{ext}^{CL}$, or applying a CCN retrieval algorithm that requires multiple assumptions, future analyses should be framed in terms of the ambient aerosol extinction

coefficient. This approach is adopted in an accompanying paper that documents the assessment of ACI over the eastern Atlantic Ocean in the Department of Energy's Energy Exascale Energy System Model (E3SM) using the dataset analyzed here (Zheng et al., 2025).

The weak values of precipitation/POP susceptibility should be interpreted as the lack of satellite data to show a significant influence of aerosols on the occurrence of precipitation. Being cognizant that the results could be dependent on the precipitation threshold

applied in our study (-15 dBZ), we also repeated the precipitation susceptibility estimate using a more stringent definition by classifying precipitating samples as those with $Z_{max}$ greater than -7 dBZ. The use of a higher precipitation threshold (not shown) did not qualitatively change the susceptibility variability of the map in Fig. 8a, yet, the number of grids with statistically insignificant values substantially increased. Given the challenges of quantifying precipitation rate, especially for stratocumulus clouds, the use of airborne observations would be necessary to complement this satellite study with the use of precipitation rate

retrievals, which will allow for more direct estimates of precipitation susceptibility.

Note that unlike previous assessments, but similar to Gryspeerdt et al. (2016), we explicitly partition the cloud adjustments into the $N_d$ modulation of LWP and precipitation (POP), and the aerosol-cloud modulation of aerosols ($\sigma_{ext}^{CL}$) on $N_d$. This method is more physically sound than directly calculating the effect of aerosol via aerosol-LWP and aerosol-precipitation regressions, because it takes into account the control variable ($N_d$) that mediates changes of aerosol in other cloud properties. Moreover, the

335 partition applied here yields a more stringent condition for evaluating the significance of the cloud susceptibility as the requirement is that two regressions are required to produce meaningful values. Another rapid cloud adjustment commonly simulated by models and monitored with satellite observations is the lifetime effect, generally represented by changes in cloud (area) fraction as a function of aerosol concentration. This cloud fraction susceptibility is not reported in this study because cloud retrievals ($N_d$ and LWP) are filtered using cloud fraction, with a threshold that directly affects the regression between CF and $N_d$. For example,

Painemal et al. (2020) show a dramatic decrease in the CF–$N_d$ slope when $N_d$ values estimated in partially broken scenes are removed from the analysis. With optical retrieval biases sensitive to the type of cloud scene and sub-pixel scale cloud coverage, disentangling the physical signature from systematic biases in the CF–$N_d$ relationship will make it difficult to determine the usability of such analysis using cloud observations from passive sensors.



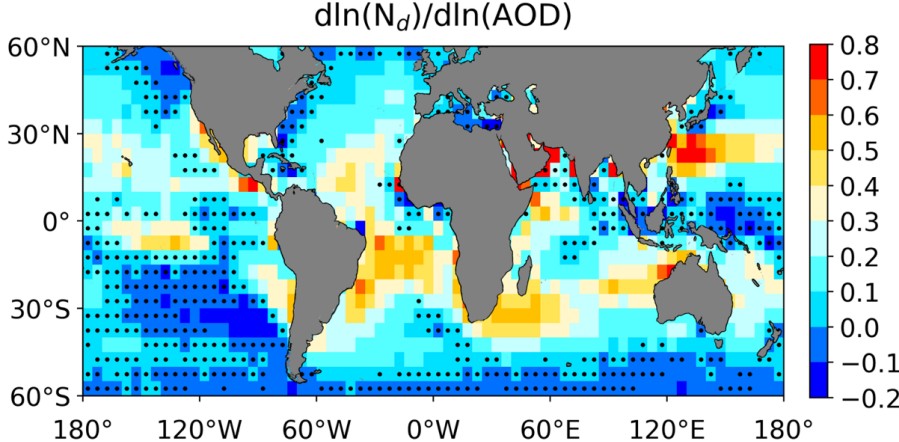

**Figure 9: Gridded maps of ACI index dln ($N_d$)/ dln ($AOD$), estimated from MODIS Science Team Meeting Level 3 daily retrievals.**

## 5 Summary and final remarks

We computed aerosol-cloud interactions and cloud adjustments over the global ocean by combining aerosol retrievals from CALIOP-S, cloud properties from MODIS (CERES algorithm), and precipitation occurrence from CloudSat. This is the first global assessment, to the best of our knowledge, that relies on vertically-resolved aerosol retrievals that are vertically matched with the location of the cloud layer. Here we expand a previous study (Painemal et al., 2020) by considering most of the A-train record (2006-2017) and including the extratropical ocean. Moreover, we also incorporate estimates of liquid water path and precipitation

susceptibilities due to aerosols to the analysis.

We corroborate that aerosol optical depth inadequately represents aerosols for the study of aerosol-cloud interactions in marine low clouds. More specifically, we found that AOD shows a negligible variation with cloud microphysics in the extra-tropics despite a strong correlation between cloud-layer aerosol extinction coefficient and $N_d$. We also found that the S-shape variations of $N_d$ with AOD reported in numerous studies and reproduced here may not fully represent the physical processes governing cloud variability.

This is because the S-shape is not replicated by the cloud-level aerosol extinction coefficient analysis presented here (Section 3.1) nor by airborne studies over the ocean. The limitation of using of AOD as an aerosol proxy for ACI is particularly manifested for values of AOD less than 0.1 and greater than 0.25, ranges for which $N_d$ minimally varies with AOD. This finding indicates that the lack of sensitivity of $N_d$ to AOD for typical aerosol loading over the ocean, is not indicative of the microphysical processes. We note that this conclusion is valid for the AOD magnitudes analyzed here, and it does not rule out other microphysical behaviors in

more polluted conditions. Indeed, thermodynamically-driven saturation of $N_d$ with aerosol loading over heavily polluted environments, especially over land, has been observed in several field studies (e.g. Ramanathan, et al., 2001).

The ACI metrics derived from combining CALIOP-S aerosol extinction coefficient, vertically collocated with the cloud layer, and MODIS products reveal regions with high sensitivity of clouds to changes in their $N_d$ due to aerosols. These areas include the stratocumulus cloud regimes off the west coast of the continents, the Southern Ocean, and the extratropical Atlantic and Pacific

Oceans in the Northern Hemisphere. Observing strong ACI in stratocumulus cloud regimes appear to be consistent with their proximity to the source of continental aerosols, in a domain where the atmospheric boundary layer is relatively well-mixed.



However, it is somewhat surprising that the Southern Ocean, arguably the most pristine region on Earth, also witnesses high values of ACI, and relatively high $N_d$ relative to other regions over the remote ocean with similar aerosol concentrations. This feature is possibly explained by strong boundary layer turbulence forced by synoptic variability, especially in the postfrontal sector (e.g.
Lang et al., 2021), combined with CCN activation-efficient aerosols with biogenic and sea-spray origins, especially in the summertime (Humphries et al., 2021).

In terms of LWP susceptibilities, this is consistently negative in subtropical and extratropical regions, that is, LWP decreases with $N_d$. This decrease in LWP, also observed in other studies (e.g. Qiu et al., 2024), is generally interpreted as the drying effect of cloud top entrainment, which is enhanced with increasing $N_d$. In contrast, positive LWP-$N_d$ slopes in the tropical ocean yield a
modest LWP susceptibility because ACI is small and insignificant. This analysis also shows that the inverted-V shape in the LWP-$N_d$ relationship is generally the consequence of spatial variability, which becomes more apparent when the calculation spatial domain is excessively large (Goren et al., 2025). Precipitation (POP) susceptibility, on the other hand, is also negative and consistent with the idea that aerosol suppresses precipitation. However, the magnitudes are only significant in narrow coastal areas in the eastern Pacific and southeast Atlantic. This is possibly related to the relatively small rain rates in these stratocumulus clouds,
making them more susceptible to changes in their precipitation frequency than regions in the extratropics with more significant rain rates.

With the successful launch of the Earth Clouds, Aerosols and Radiation Explorer mission (EarthCARE; Wehr et al., 2023) in May 2024, the EarthCARE sensors will enable assessing aerosol-cloud interactions with products that will largely expand the capabilities of CALIPSO and CloudSat. For example, the improved sensitivity of the EarthCARE Cloud Profiling radar will
enhance the detection of clouds relative to CloudSat, detecting clouds as low as 600 m. In addition, the EarthCARE Atmospheric Lidar (ATLID), being a high spectral resolution lidar, will provide direct observations of aerosol extinction coefficient and refined aerosol typing classification. Because of the sampling and collocation constraints in our study (which includes more than 11 years of A-Train observations), multiple years of EarthCARE observations will be required to replicate the statistical robustness of our analysis. Alternatively, efforts for expanding the lidar-cloud record through the homogenization of CloudSat, CALIPSO, and
EarthCARE products will be necessary to corroborate and expand the findings of this study.

**Competing Interests**: The authors declare no competing interests.

**Author Contributions**. DP and YF developed the research concept with contributions from ZL. ZL and DP conducted the research and wrote the manuscript, with contributions from YF and XZ.

**Data and code availability.** The 25-km merged dataset used in the analysis is currently being prepared for making it available in a NASA repository. The data will be available before the manuscript is accepted for publication.

**Acknowledgement**

This research was funded by the CloudSat and CALIPSO Science Team Recompete Program under the Science Mission Directorate of NASA (NNH21ZDA001N-CCST). Y.F. also acknowledges the support of the Atmospheric System Research program, funded by the U.S. Department of Energy (DOE), Office of Science, Office of Biological and Environmental Research. The work at Argonne National Laboratory was supported by the U.S. DOE Office of Science under contract DE-AC02-06CH11357.



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
