# Peer review of "Progress in the quantification of aerosol-cloud interactions estimated from the CALIPSO-CloudSat-Aqua/MODIS record"

_EGUsphere, 2025_

## Referee Comment (RC1)

Li et al. combine multiple spaceborne A-Train sensors to investigate aerosol–cloud interactions (ACI). They assess the susceptibility of cloud droplet number concentration (Nd) and liquid water path (LWP), derived from MODIS, to aerosol extinction coefficients retrieved at cloud level from CALIPSO. They also examine the relationship between aerosol extinction coefficients and the occurrence of precipitation using CloudSat observations. The effort to relate aerosol properties at cloud level with cloud characteristics on a global scale is scientifically significant. However, the implementation of this study contains several methodological shortcomings that have been extensively documented in previous literature and should be carefully considered when correlating optical aerosol properties with cloud parameters.

In the introduction, the authors appropriately highlight several known limitations of using AOD or AI in ACI studies:

- **Lines 41–43:** This is because AOD (or aerosol index) does not uniquely represent aerosol concentration or CCN concentration, as variations in aerosol composition, particle size distribution, and optical properties can yield the same AOD for different aerosol concentrations.
- **Lines 43–44:** A second limitation is the inability to disentangle the contributions of different aerosol layers to the total AOD, which prevents any meaningful vertical collocation between aerosol and cloud layer.
- Lines 58–60: Moreover, enhanced aerosol swelling, cloud contamination, and three-dimensional radiative effects can affect the collocated satellite AOD pixels near cloud edges (Varnai and Marshak, 2009).

Given these acknowledgments and the manuscript's title, one would expect substantive progress in addressing these limitations. However, only the issue of vertical collocation has been considered, following their previous work (Paneimal et al., 2020). Other important sources of uncertainty, such as variations in aerosol type and size distribution and the influence of aerosol hygroscopic growth, are equally relevant for CALIPSO-derived extinction profiles. These factors are either neglected, deemed insignificant without sufficient justification, or, surprisingly, suggested to be not important for future ACI studies in the discussion section.

I have several major concerns, primarily regarding the aerosol and cloud sampling criteria employed in this analysis. These include the inappropriate inclusion of precipitating clouds in the computation of Nd susceptibility, the use of aerosol properties from highly humid regions adjacent to clouds, the restriction to broken-cloud 25 km X 25 km scenes for estimating LWP susceptibility, and the fine spatial aggregation applied in the analysis. Each

of these issues could significantly affect the derived sensitivities and should be carefully revisited. Addressing these points is essential for the manuscript to substantiate its claim of advancing the assessment of aerosol–cloud interactions.

**Major Comments:**

1. Line 145: The authors limit the 25 by 25 km cloud fraction (CF) to 90% to exclude cases where aerosols are fully embedded within cloudy regions, on the premise that such situations are affected by aerosol swelling due to hygroscopic growth at high relative humidity (RH). However, this filtering does not adequately ensure that hygroscopic growth is properly accounted for. Aerosol retrievals in direct contact with cloudy pixels (likely cloud-contaminated pixels) can still be significantly influenced by hygroscopic growth effects, irrespective of CF. As demonstrated in Christensen et al. (2017), this can lead to artificially enhanced correlations between Nd and AOD or AI. Since the cloud-level aerosol extinction coefficients are considered in the present manuscript, where the RH effect is likely significant, the derived susceptibilities may be biased.

I recommend redoing the calculations after omitting aerosol retrievals in pixels directly adjacent to cloudy columns irrespective of total CF. This will also address another issue in computing dlnLWP/dlnNd (see next paragraph). This approach has been adopted in several recent ACI studies using satellite-derived AI to estimate Nd susceptibility (e.g., Jia et al., 2022). Alternatively, aerosol retrievals can be filtered using an RH threshold (e.g., only including retrievals where RH < 70-80%), within which hygroscopic growth is limited for both continental and marine aerosol types. RH values can be obtained from the operational CALIPSO product (which includes interpolated meteorological parameters) or directly from reanalysis datasets such as ERA5 or MERRA-2. This is a fundamental consideration in satellite-based ACI studies and should not be overlooked, particularly in a study aiming to advance current estimates of Nd susceptibility.

Furthermore, the decision to omit cloud retrievals with CF > 90% (within 25  $\times$  25 km scenes) when computing dlnLWP/dlnNd is not justified. Both LWP and Nd are derived from MODIS cloud retrievals, which tend to be more reliable in overcast cloud fields due to their higher spatial homogeneity. Such conditions better satisfy the plane-parallel cloud approximation, and consequently, three-dimensional radiative effects are minimized (Zhang and Platnick, 2011). I recommend removing the CF filtering from Nd-LWP susceptibility calculations.

- 2. Lines 236–237: The authors state, "Indeed, global ACI for non-precipitating (Zmax < –15 dBZ) and precipitating (Zmax > –15 dBZ) segments is 0.13 and 0.08, respectively." It is unclear how this information can be inferred from Fig. 5. I assume that the authors averaged the ACI indices over grid points with the minimum or maximum probability of precipitation (POP). If this interpretation is correct, further clarification is necessary on how this separation was implemented and statistically represented in the figure. Based on this assumption, I have an additional related comment below.
- 3. Another fundamental issue not addressed in this study is the inclusion of precipitating clouds in the calculation of the ACI index or Nd susceptibility, which leads to two key issues. First, precipitating clouds introduce significant uncertainty in Nd retrievals, as the assumption of adiabaticity no longer holds. Second, collision-coalescence reduces Nd independent of aerosol loading, thereby distorting the aerosol–cloud relationship. The inclusion of precipitating scenes can lead to a non-causal positive bias in Nd susceptibility of approximately 21% (Jia et al., 2022). Since the authors already utilize CloudSat observations to identify precipitating clouds, it would be straightforward to exclude precipitating clouds from the analysis and recompute Nd susceptibility accordingly.
- 4. Since the authors use LWP and Nd from MODIS following a similar approach to previous studies (e.g., Gryspeerdt et al., 2019), the primary differences between their results and those in the literature appear to stem from the finer aggregation scale (25 km × 25 km instead of 100 km × 100 km) and the exclusion of pixels with CF > 90%. One concern here is the use of such a fine grid size. A 25 km × 25 km domain may not be sufficiently large to capture the structural or morphological variability within cloud systems over oceans. While cloud-top Nd tends to be relatively homogeneous in non-precipitating clouds, as it is primarily governed by the initially activated CCN population, the situation is different for LWP. Within a cloud, LWP typically peaks in the core regions and decreases toward the periphery, leading to substantial intra-cloud heterogeneity. This variability becomes even more pronounced in precipitating clouds. So, for similar Nd, we can have two different LWP, because of the cloud morphology, not directly because of aerosols. It is unclear how these in-cloud variations are accounted for in the current analysis, and clarification on this point is necessary to assess the robustness of the derived susceptibilities.

5. **Line 319**: The authors state that "future analyses should be framed in terms of the ambient aerosol extinction coefficient." It is unclear how this recommendation is justified, given that aerosol hygroscopic growth is known to bias Nd susceptibility estimates. Numerous previous studies have recognized and explicitly accounted for this effect (e.g., Christensen et al., 2017; Hasekamp et al., 2019; Jia et al., 2022; Quaas et al., 2020). The authors should clarify the rationale behind this suggestion.

**Minor comments:**

- 6. Line 26: "Observational estimates ..." instead of "Estimates"?
- 7. Lines 48-49: Do you mean the "updraft limited regime" (Reutter et al., 2009)?
- 8. Line 64: Citing the authors: "Regrettably, the application of spaceborne lidar observations to the ACI computation is still surprisingly lacking." This is not entirely true. Alexandri et al. (2024) combined CALIPSO-derived CCN concentrations with Nd from geostationary observations in a sophisticated cloud-by-cloud framework using an advanced cloud tracking and matching algorithm.
- 9. Line 106: Which wavelength was used for the effective radius and why? Did the authors apply the condensation rate temperature correction based on Gryspeerdt et al. (2019) when calculating Nd?
- 10. Which correlation coefficient is shown in Figures 2 and 3? Please mention it in the caption. I recommend the pearson's correlation coefficient. If the authors prefer spearman, please provide the figures with pearson's correlation coefficient in the supplementary.
- 11. Figure 4: How do the authors interpret negative dlnNd/dlnEXT
- 12. Line 262: dlnLWP/dlnNd is also affected by sampling bias due to missing cloud properties in MODIS as a result of retrieval failure, particularly the positive dlnLWP/dlnNd response (Choudhury and Goren, 2025).
- 13. I suggest the authors provide a supplementary figure showing dlnNd/dln(EXTsurface) and dlnNd/dln(AOD)?
- 14. A general observation from Figures 4 and 9 is low or negative ACI index over pristine oceans. Can the authors comment on why this could happen in both CALIPSO and MODIS retrievals?

**References:**

Alexandri, F., Müller, F., Choudhury, G., Achtert, P., Seelig, T., and Tesche, M.: A cloud-by-cloud approach for studying aerosol–cloud interaction in satellite observations, Atmos. Meas. Tech., 17, 1739–1757, https://doi.org/10.5194/amt-17-1739-2024, 2024.

Christensen, M. W., Neubauer, D., Poulsen, C. A., Thomas, G. E., McGarragh, G. R., Povey, A. C., Proud, S. R., and Grainger, R. G.: Unveiling aerosol–cloud interactions – Part 1: Cloud contamination in satellite products enhances the aerosol indirect forcing estimate, Atmos. Chem. Phys., 17, 13151–13164, https://doi.org/10.5194/acp-17-13151-2017, 2017.

Choudhury, G., & Goren, T. (2025). Sampling bias from satellite retrieval failures of cloud properties and its implications for aerosol-cloud interactions. Geophysical Research Letters, 52, e2025GL115429. https://doi.org/10.1029/2025GL115429

Gryspeerdt, E., Goren, T., Sourdeval, O., Quaas, J., Mülmenstädt, J., Dipu, S., Unglaub, C., Gettelman, A., and Christensen, M.: Constraining the aerosol influence on cloud liquid water path, Atmos. Chem. Phys., 19, 5331–5347, https://doi.org/10.5194/acp-19-5331-2019, 2019.

Painemal, D., Chang, F.-L., Ferrare, R., Burton, S., Li, Z., Smith Jr., W. L., Minnis, P., Feng, Y., and Clayton, M.: Reducing uncertainties in satellite estimates of aerosol–cloud interactions over the subtropical ocean by integrating vertically resolved aerosol observations, Atmos. Chem. Phys., 20, 7167–7177, doi:10.5194/acp-20-7167-2020, 2020.

Hasekamp, O.P., Gryspeerdt, E. & Quaas, J. Analysis of polarimetric satellite measurements suggests stronger cooling due to aerosol-cloud interactions. Nat Commun 10, 5405 (2019). https://doi.org/10.1038/s41467-019-13372-2

Jia, H., Quaas, J., Gryspeerdt, E., Böhm, C., and Sourdeval, O.: Addressing the difficulties in quantifying droplet number response to aerosol from satellite observations, Atmos. Chem. Phys., 22, 7353–7372, https://doi.org/10.5194/acp-22-7353-2022, 2022.

Quaas, J., Arola, A., Cairns, B., Christensen, M., Deneke, H., Ekman, A. M. L., Feingold, G., Fridlind, A., Gryspeerdt, E., Hasekamp, O., Li, Z., Lipponen, A., Ma, P.-L., Mülmenstädt, J., Nenes, A., Penner, J. E., Rosenfeld, D., Schrödner, R., Sinclair, K., Sourdeval, O., Stier, P., Tesche, M., van Diedenhoven, B., and Wendisch, M.: Constraining the Twomey effect from satellite observations: issues and perspectives, Atmos. Chem. Phys., 20, 15079–15099, https://doi.org/10.5194/acp-20-15079-2020, 2020.

Reutter, P., Su, H., Trentmann, J., Simmel, M., Rose, D., Gunthe, S. S., Wernli, H., Andreae, M. O., and Pöschl, U.: Aerosol- and updraft-limited regimes of cloud droplet formation: influence of particle number, size and hygroscopicity on the activation of cloud condensation nuclei (CCN), Atmos. Chem. Phys., 9, 7067–7080, https://doi.org/10.5194/acp-9-7067-2009, 2009.

Zhang, Z., and S. Platnick (2011), An assessment of differences between cloud effective particle radius retrievals for marine water clouds from three MODIS spectral bands, J. Geophys. Res., 116, D20215, doi:10.1029/2011JD016216.

---

## Referee Comment (RC2)

This work estimates various cloud susceptibilities, including Nd-to-aerosol, LWP-to-Nd and precipitation-to-Nd, using long-term synergistic dataset of CALIPSO-CloudSat-Aqua/MODIS. A specific focus is on how the vertical co-location of cloud and aerosol layer affects the ACI estimates. This produced dataset that the authors will make it publicly available, would be very useful for the ACI community.

The manuscript is overall well-written and the methodology is sound. However, some conclusions appear somewhat too strong, and a few statements require clarification (see my detailed comments below). Despite these points, I believe this could be an interesting study for ACP once all concerns listed are addressed.

**Major comments:**

Introduction:

> I feel many important references were not mentioned in this study. Most points discussed in the Introduction have been already well-documented in previous review papers, e.g., https://doi.org/10.5194/acp-20-15079-2020 and more recent https://doi.org/10.1029/2022RG000799 and the references therein. Would be nice to acknowledge previous work.

> For the effect of retrieval bias, I don't think (Varnai and Marshak, 2009) really touched the effect on ACI, instead more about AOD error. A detailed investigation can refer to https://doi.org/10.5194/acp-22-7353-2022, which may be more relevant here. Regarding the aerosol retrieval issue in low aerosol conditions, a reference should be provided (see the discussion about this in above two review papers).

Figure. 2: Similar plots but showing sample size would be help here. Also, it's interesting to see negative signals in some regions, particularly in (b) this seems to be more visible in regions with strong precipitation. Any explanation about this? As I can understand these plots were for all clouds; would be interesting to look at non-precipitating clouds only. Similarly, for L233-237: The easiest way to investigate the effect of precipitation on ACI is making the similar plots as Fig. 4 but distinguishing non-precipitating and precipitating clouds

L213: maybe an explanation on Spearman correlation would help. It's confusing that the data in fig. 3a apparently are not concentrated around the regression line, but r_s are mostly larger than 0.95 and even being 1. Please clarify.

L216-219 (also the argument on 'S-shape' in abstract): I'm not sure how much I can be convinced by this statement.

- I feel the reason why we didn't see a clear 'flat curve' in high $\sigma$ is the insufficient samples there; binning data into same sample-size bins induces a weak representativity where data are sparse. Even with sparse data, we still see the saturated Nd when $\sigma$ starts going beyond 0.2-0.3. Thus, the analysis here is not sufficient to demonstrate the S-shape is non-physical.

- Even using boundary-layer SO4 (closer to $\sigma$ here), the sigmoid shape is still quite clear (fig. 1b in https://doi.org/10.5194/acp-23-4115-2023). A recent study further provided the observational evidence for this sigmoid curve based on long-term trends (https://doi.org/10.1038/s41558-023-01775-5). These should be discussed.
- I'd suggest formulating it in a way that the non-lineaer behavior reported in earlier studies tends to be less pronounce when using cloud-base extinction than column AOD, instead of saying it's non-physical as the results presented cannot justify this strong statement.

L260-261: This is an interesting point. The authors could demonstrate this even clearer by making a panel (b) for Fig 7 but lumping all global data together.

**Minor comment:**

Since only the vertical co-location of cloud and aerosol layers is studied, the term 'Progress' in the title seems too broad and gives the impression of a review-like paper. I suggest removing it.

L26: Estimates -> Observational estimates

This work largely follows Painemal et al. (2020). The importance of using vertically collocated aerosol has been already well-justified. Would be good to explain what new message one could get beyond the existing literature.

L68: 'shortcomings': Since the text so far only highlights the benefits, it might be helpful to flag what shortcomings the readers can expect next.

' metrics of cloud susceptibilities of ACI' and 'cloud susceptibility' are the same, aren't they?

L80-82: this sentence is hard to read. Please explain what 'This choice of CALIPSO-based dataset responds to limitations of the standard CALIPSO product' means

L93: I personally think CTH is a better term than ZT for cloud top hight, which has been widely used. Would be easier for readers

L99: 'height': is it cloud top height?

Eq 1: Though the authors referred to (Albertcht et., 1990), it's good to provide the full formulation here along with all parameter values need in the calculation so that people can easily follow.

112-113: How it can categorize the low-cloud precipitation rate is not clear. I guess the authors put a 'raining' flag if Zmax?-15, otherwise 'non-raining', right?

L124: Could you clarify what you mean by "the closest CloudSat CPR pixels to the 25-km line"? What exactly does the 25 km line refer to here?

L134-136: it's a bit unclear if the cloud-top height is from MODIS or CALIOP as stated earlier? To match and 1x1 modis pixels I'd assume it's from MODIS right?

L140: the threshold is generally set to 4; could you explain why 2 is used here? Does it mean more optically thin clouds are included in this study?

Eq3: Only data with CF>80% are analysed. In this case, S_CF cannot reflect the real effect. would be nice to mention this limitation here though it appears quite later in the paper

L207: 'is constrained using AOD' what does this mean? Is $\sigma$ vertically integrated into the value of AOD?

Fig.7: It would be easier to follow if the authors marked these 4 regions in Fig 6.

l261-262: I think the story in Arola et al. (2022) is quite different to the argument here. They attributed the invert-V to retravel errors. Citing this paper here seems a bit confusing unless the authors make this clear.

L266: It's Intuitive that the product of S_nd_lwp (fig. 6a) and ACI (Fig. 4a) should be negative as their signs are opposite, especially in Tropics; so it's kinda surprising that it turns to be negligible. Could the authors explain this a bit more?

L298: I'd avoid words like "novel" or "new," or anything implying the study is the first to show a particular conclusion. A more neutral phrasing would work better.

It would be easier for readers to follow the results if Fig. 4 were placed as a separate panel within Fig. 9; so that readers would not need to scroll back and forth. And why does the ACI(based on AOD) index appear to be negative here? It's overall positive in previous studies. Would be good to discuss.

L301-303: The authors stressed a lot on the difference in ACI between $\sigma$ and AOD; but it's very important to mention here that in the end we care about anrgropogenic perturbation of Nd and forcing which also relies on PI-PD change in the utilized proxy, not the simple slope (https://doi.org/10.1029/2022RG000799).

L311-312: the use of AOD doesn't misguide the modelers as long as they are looking at AOD as well. I think this sentence can be dropped.

L311: I find the phrase "unlike previous assessments, but similar to Gryspeerdt et al. (2016)" a bit confusing. Gryspeerdt et al. (2016) is also a 'previous' study, so it might help to clarify what you mean here. For example, do you refer to a specific group of 'previous assessments' using a different methodology?

---

## Author Comment (AC1)

**Reply to Reviewer # 1**

We truly appreciate the reviewer's report and his/her comments that rightly challenge our analysis and result interpretation. Our responses will serve as a guideline for revising our manuscript. His/her specific comments are addressed below (highlighted in blue).

"Given these acknowledgments and the manuscript's title, one would expect substantive progress in addressing these limitations. However, only the issue of vertical collocation has been considered, following their previous work (Paneimal et al., 2020). Other important sources of uncertainty, such as variations in aerosol type and size distribution and the influence of aerosol hygroscopic growth, are equally relevant for CALIPSO-derived extinction profiles. These factors are either neglected, deemed insignificant without sufficient justification, or, surprisingly, suggested to be not important for future ACI studies in the discussion section."

Thanks for pointing out that the introduction did not meet the reader's expectations about the aspects that the manuscript actually addresses and those that remain unanswered due to the limitations of our observational dataset. In our search for conciseness, we did not provide enough necessary details the rationale behind a given analysis or conclusion. In the revised manuscript, we will describe more clearly what is specifically addressed by our study and will be more precise about our assumptions and uncertainties of the analysis. In addition, we include a lidar ratio analysis to address the role of aerosol type and particle size in the analysis. The general reviewer's concerns are more specifically addressed in the following. Regarding the title, we will be revising the title of the manuscript to: "Advancing the quantification of aerosol-cloud interactions with the use of the CALIPSO-CloudSat-Aqua/MODIS record".

"I have several major concerns, primarily regarding the aerosol and cloud sampling criteria employed in this analysis. These include the inappropriate inclusion of precipitating clouds in the computation of Nd susceptibility, the use of aerosol properties from highly humid regions adjacent to clouds, the restriction to broken-cloud 25 km X 25 km scenes for estimating LWP susceptibility, and the fine spatial aggregation applied in the analysis. Each of these issues could significantly affect the derived sensitivities and should be carefully revisited. Addressing these points is essential for the manuscript to substantiate its claim of advancing the assessment of aerosol–cloud interactions."

We are grateful to the reviewer for bringing these points up. All these concerns are valid and are responded in detail below.

Major Comments:

1.      "The authors limit the 25 by 25 km cloud fraction (CF) to 90% to exclude cases where aerosols are fully embedded within cloudy regions, on the premise that such situations are affected by aerosol swelling due to hygroscopic growth at high relative humidity (RH). However, this filtering does not adequately ensure that hygroscopic growth is properly accounted for. Aerosol retrievals in direct contact with cloudy pixels (likely cloud-contaminated pixels) can still be significantly influenced by hygroscopic growth effects, irrespective of CF. As demonstrated in Christensen et al. (2017), this can lead to artificially enhanced correlations between Nd and AOD or AI. Since the cloud-level aerosol extinction coefficients are considered in the present manuscript, where the RH effect is likely significant, the derived susceptibilities may be biased."

The reviewer's points are highly pertinent to our study. There are 2 aspects of aerosol hygroscopicity that need further discussion.

a)  Variable effect of hygroscopicity attributed to the proximity of the aerosol pixel to clouds: This is our primary concern because, as the reviewer is aware, studies have shown that the dependence of AOD on cloud fraction (CF) is primarily the effects of multiple artifacts in the aerosol retrievals, rather than a physical signature of cloud adjustment (e.g. Varnai et al.). So, our data filtering was primarily intended to minimize the sensitivity of aerosol retrievals to cloud coverage. As explained in the manuscript, the effect of clouds on aerosol is not only the influence of hygroscopicity but also the substantial effect of aerosol-cloud misclassification, and 3-D radiative transfer effects. Key advantages of CALIOP include: insensitivity to 3-D radiative effects, and an improved aerosol and cloud identification relative to passive imagers like MODIS. Given the advantages of CALIOP, we conclude that the analysis of Christensen et al. (2017) is only representative of MODIS AOD and similar products derived from passive sensors. A way to visualize the effect of clouds in AOD retrievals is by analyzing the relationship between CF and AOD. Fig R1a illustrates this relationship. First, our CALIOP-based AOD shows a modest increase with CF, which only becomes severe for CF>0.95, that is, when the CALIOP pixels are surrounded by clouds. Because we remove samples with MODIS CF>0.9, we can effectively remove CALIOP grids more affected by aerosol swelling due to clouds. In a similar manner, filtering our cloud retrievals (N_d, FigR1a, red line) minimizes the Nd dependence on CF (Fig. R1b). All in all, the final filtering of both Nd and CALIPSO-SODA AOD (Fig R1b, red circles) yield a much weaker slope relative to data without filtering. This shows that our method removes multiple effects and artifacts that could conspire to enhance ACI. Lastly, we would like to remind the reviewer that this CALIOP aerosol retrievals are only used if the corresponding 5km along-track CALIOP grid is cloud free.

[Figure]

Figure R 1: Figure adapted from Painemal et al. 2020 (Fig. 2a). a) relationship between segment CF from MODIS and: all-sky Nd (without filtering), cloudy Nd (cloud fraction >90% within a 5 pixel x 5 pixel box), and AOD averaged for 5-km cloud-free CALIOP grids. b) relationship between AOD and Nd without filtering (black) and after applying CF filtering (red).

b) Hygroscopicity as a function of the ambient relative humidity (RH). The figure below depicts the mean relative humidity at around 800 m (925 hPa) from MERRA-2. Notably, RH exceeds 80 % (0.8) for most of the oceanic regions. Smaller RH values are observed over the eastern Pacific and Atlantic, because the inversion height in those regions is below 925 hPa (in addition to the potential misrepresentation of the boundary layer height in the model). The figure also indicates that the range of RH variability is somewhat constrained to a narrow range. In other words, in the context of the reviewer's comment, regional changes in AOD are primarily driven by the aerosol type and their specific hygroscopicity rather than variability in RH. Even if RH modulates the absolute value of AOD and extinction coefficient, this does not necessarily translate to biases in ACI. Unfortunately, we do not count on the dataset to address this science question (see our replies below).

[Figure]

Figure R2: Annual mean relative humidity at 925 hPa (~800 m). Values range between 0.3 and 0.95 (35% and 95%).

"I recommend redoing the calculations after omitting aerosol retrievals in pixels directly adjacent to cloudy columns irrespective of total CF. This will also address another issue in computing dlnLWP/dlnNd (see next paragraph). This approach has been adopted in several recent ACI studies using satellite-derived AI to estimate Nd susceptibility (e.g., Jia et al., 2022).

We show in our reply above that: a) our method does remove effects of aerosol swelling near the vicinity of clouds, and b) Multiple artifacts reported in the literature regarding multiple cloud effects on AOD derived from passive sensors are substantially ameliorated when using CALIOP. Regarding the latter, Yang et al. (2014) conclude based on CALIOP data: "This result suggests that systematic changes in the near-cloud transition zone are real but somewhat weaker than previously reported and that understanding the statistics of near-cloud aerosol properties requires a consideration of changes in cloud fraction." The conclusions in Yang et al. are consistent with our assertion about the quality of the CALIPSO retrievals and our data filtering.

Reference:

Yang, W., A. Marshak, T. Várnai, and R. Wood (2015), CALIPSO observations of near-cloud aerosol properties as a function of cloud fraction, *Geophys. Res. Lett.*, 41, 9150–9157, doi:10.1002/2014GL061896.

Alternatively, aerosol retrievals can be filtered using an RH threshold (e.g., only including retrievals where RH < 70-80%), within which hygroscopic growth is limited for both continental and marine aerosol types. RH values can be obtained from the operational CALIPSO product (which includes interpolated meteorological parameters) or directly from reanalysis datasets such as ERA5 or MERRA-2. This is a fundamental consideration in satellite-based ACI studies and should not be overlooked, particularly in a study aiming to advance current estimates of Nd susceptibility."

As shown in Fig. R2, the high RH in the boundary layer makes the reviewer suggestion challenging to implement. A second question to address is to account for aerosol swelling, that is, converting, aerosol extinctions from ambient RH to dry values (RH< 50%). This is generally done by assuming a parameterization that is a function of RH (e.g. Gasso et al., 2000; Zieger et al., 2013). Zieger et al. (2013) show that this scattering enhancement factor can vary significantly depending on the air mass and aerosol composition. It might be possible to characterize the aerosol types using CALIPSO aerosol classification, but we argue that this typing does not provide sufficient information nor a consistent optical characterization for our study (e.g. Li et al., 2022). Because the scattering enhancement factor parameterization is sensitive to the aerosol mass and we do not count on a reliable way to characterize the aerosol hygroscopicity, we decided not to apply any correction that could introduce more uncertainties.

Reference:
Zieger, P., Fierz-Schmidhauser, R., Weingartner, E., and Baltensperger, U.: Effects of relative humidity on aerosol light scattering: results from different European sites, Atmos. Chem. Phys., 13, 10609–10631, https://doi.org/10.5194/acp-13-10609-2013, 2013.

S. Gassó, D. A. Hegg, D. S. Covert, D. Collins, K. J. Noone, E. Öström, B. Schmid, P. B. Russell, J. M. Livingston, P. A. Durkee & H. Jonsson (2000) Influence of humidity on the aerosol scattering coefficient and its effect on the upwelling radiance during ACE-2, Tellus B: Chemical and Physical Meteorology, 52:2, 546-567, DOI: 10.3402/tellusb.v52i2.16657

Li, Z., Painemal, D., Schuster, G., Clayton, M., Ferrare, R., Vaughan, M., Josset, D., Kar, J., and Trepte, C.: Assessment of tropospheric CALIPSO Version 4.2 aerosol types over the ocean using independent CALIPSO–SODA lidar ratios, Atmos. Meas. Tech., 15, 2745–2766, https://doi.org/10.5194/amt-15-2745-2022, 2022.

"Furthermore, the decision to omit cloud retrievals with CF > 90% (within 25 × 25 km scenes) when computing dlnLWP/dlnNd is not justified. Both LWP and Nd are derived from MODIS cloud retrievals, which tend to be more reliable in overcast cloud fields due to their higher spatial homogeneity. Such conditions better satisfy the plane-parallel cloud approximation, and consequently, three-dimensional radiative effects are minimized (Zhang and Platnick, 2011). I recommend removing the CF filtering from Nd-LWP susceptibility calculations."

The reviewer is correct in that removing fully overcast scenes in the context of quantifying dlnLWP/dlnNd is unjustified. The single reason why we adopted this filtering was for consistency in the methodology because ACI was calculated using the same manner. The new figure is included below. These new maps are nearly identical to their counterparts in the original manuscript.

[Figure]

Figure R3: Gridded maps of (a) susceptibility of LWP to $N_d$ or $S_{LWP}^{Nd} = \frac{\partial \ln(LWP)}{\partial \ln(N_d)}$; and (b) overall LWP susceptibility to aerosols estimated as $S_{LWP} = S_{LWP}^{Nd} \cdot ACI$. Black dots in (a) indicate grids that are statistically indistinguishable from zero, according to a Student's t test at 95% confidence level, whereas dots in (b) represent boxes when at least one metric (ACI or $S_{LWP}^{Nd}$) is statistically indistinguishable from zero. The LWP susceptibility computation includes 25-km cloud fraction > 0.9 (90%).

2.      Lines 236–237: The authors state, "Indeed, global ACI for non-precipitating (Zmax < –15 dBZ) and precipitating (Zmax > –15 dBZ) segments is 0.13 and 0.08, respectively. " It is unclear how this information can be inferred from Fig. 5. I assume that the authors averaged the ACI indices over grid points with the minimum or maximum probability of precipitation (POP). If this interpretation is correct, further clarification is necessary on how this separation was implemented and statistically represented in the figure. Based on this assumption, I have an additional related comment below.

Indeed, the statement is unclear. POP=0 and POP>0 define the non-precipitating and precipitating observations. This is now clarified in the revised manuscript. The general question about precipitating vs non-precipitating samples is addressed in the following.

3.	Another fundamental issue not addressed in this study is the inclusion of precipitating clouds in the calculation of the ACI index or Nd susceptibility, which leads to two key issues. First, precipitating clouds introduce significant uncertainty in Nd retrievals, as the assumption of adiabaticity no longer holds. Second, collision-coalescence reduces Nd independent of aerosol loading, thereby distorting the aerosol–cloud relationship. The inclusion of precipitating scenes can lead to a non-causal positive bias in Nd susceptibility of approximately 21% (Jia et al., 2022). Since the authors already utilize CloudSat observations to identify precipitating clouds, it would be straightforward to exclude precipitating clouds from the analysis and recompute Nd susceptibility accordingly.

Correct, it is expected that precipitation will affect the magnitude of the slopes (see our previous response). During the early stage of the analysis, our goal was to provide ACI maps that could be easily compared against other datasets or model outputs. This is the primary reason why we did not separate precipitating from non-precipitating samples. However, we see the value in implementing the reviewer's suggestion and we are now including computations estimated for precipitating and non-precipitating data. Figure R4 depicts statistics for precipitating samples, defined as those with probability of precipitation (POP) higher than 0.3, and non-precipitating clouds for samples with POP <0.05.

[Figure]

Figure R4: Gridded map of ACI index (d ln $(N_d)$/ d ln $(\sigma_{ext}^{CL})$ ). Black dots indicate grids that are statistically indistinguishable from zero, according to a Student's t test at 95% confidence level. a) ACI index for sampling with significant precipitation frequency (POP>0.3), and b) non-precipitating clouds (POP<0.05).

4.      Since the authors use LWP and Nd from MODIS following a similar approach to previous studies (e.g., Gryspeerdt et al., 2019), the primary differences between their results and those in the literature appear to stem from the finer aggregation scale (25 km × 25 km instead of 100 km × 100 km) and the exclusion of pixels with CF > 90%. One concern here is the use of such a fine grid size. A 25 km × 25 km domain may not be sufficiently large to capture the structural or morphological variability within cloud systems over oceans. While cloud-top Nd tends to be relatively homogeneous in non-precipitating clouds, as it is primarily governed by the initially activated CCN population, the situation is different for LWP. Within a cloud, LWP typically peaks in the core regions and decreases toward the periphery, leading to substantial intra-cloud heterogeneity. This variability becomes even more pronounced in precipitating clouds. So, for similar Nd, we can have two different LWP, because of the cloud morphology, not directly because of aerosols. It is unclear how these in-cloud variations are accounted for in the current analysis, and clarification on this point is necessary to assess the robustness of the derived susceptibilities.5.

The author raises an interesting point. However, it is unclear to us how the effect of heterogeneity might affect the computation of susceptibility. For example, while a 25 km scale might seem small, it is within the range of spatial variability of closed-cell structures (Wood and Hartmann et al., 2006, https://doi.org/10.1175/JCLI3702.1), which are dominant in subtropical and postfrontal regions. 25km is also larger than the size of shallow cumulus clouds. So, we find ourselves in the difficult situation of choosing a scale that represents the range of variability of marine low clouds, while, at the same time, a grid sufficiently small to assume that cloud-free aerosol retrievals are representative of the nearly-adjacent cloudy areas.  It is also relevant to recall that regressions are computed over 5°x5° grids. Considering all these points, we do not have sufficient arguments to change the spatial collocation of Figure 1.

5.      Line 319: The authors state that "future analyses should be framed in terms of the ambient aerosol extinction coefficient." It is unclear how this recommendation is justified, given that aerosol hygroscopic growth is known to bias Nd susceptibility estimates. Numerous previous studies have recognized and explicitly accounted for this effect (e.g., Christensen et al., 2017; Hasekamp et al., 2019; Jia et al., 2022; Quaas et al., 2020). The authors should clarify the rationale behind this suggestion.

Regarding the comment about the justification for using ambient aerosol extinction coefficient: While methods for accounting for hygroscopicity have been presented in the literature, we are convinced that their applications to satellite data have not validated with the necessary details, nor the uncertainties assessed to the point that we can fully rely on these methods. That is, we argue that unless retrieval refinements are rigorously compared against independent datasets, it is recommended to directly use the satellite retrievals.

Minor comments:
6.      Line 26: "Observational estimates …" instead of "Estimates"?

Corrected, thanks.

7.      Lines 48-49: Do you mean the "updraft limited regime" (Reutter et al., 2009)?
        Yes, we are referring to the updraft limited regime. We appreciate the reviewer for suggesting the Reutter et al. article. The article is now cited in the same section.

8.      Line 64: Citing the authors: "Regrettably, the application of spaceborne lidar observations to the ACI computation is still surprisingly lacking." This is not entirely true. Alexandri et al. (2024) combined CALIPSO-derived CCN concentrations with Nd from geostationary observations in a sophisticated cloud-by-cloud framework using an advanced cloud tracking and matching algorithm.
        We agree with the reviewer in that the sentence was inaccurate or, at least, too strong. In the revised version, we rephrased the sentence to read: "Regrettably, studies that make use of spaceborne lidar observations for ACI studies are surprisingly scarce, and global scale analyses are lacking". Alexandri et al. (2024) will be discussed in more detailed in the discussion section.

9.      Line 106: Which wavelength was used for the effective radius and why? Did the authors apply the condensation rate temperature correction based on Gryspeerdt et al. (2019) when calculating Nd?
        We used CERES-MODIS droplet effective radius derived using the 3.7 um channel. This wavelength is less sensitive to 3D radiative effects, and spatial inhomogeneities (Painemal et al., 2013 and references therein). Unlike Gryspeerdt et al (2019), we directly used an analytical formulation for estimating the adiabatic lapse rate. In other words, no corrections are needed because the derivation is directly estimated from adiabatic considerations (see Albrecht et al. 1990). It suffices to say that the adiabatic computation follows the thermodynamic equation described in Albrecht et al.

Reference:
Painemal, D., Minnis, P., and Sun-Mack, S.: The impact of horizontal heterogeneities, cloud fraction, and liquid water path on warm cloud effective radii from CERES-like Aqua MODIS retrievals, Atmos. Chem. Phys., 13, 9997–10003, https://doi.org/10.5194/acp-13-9997-2013, 2013.

10.     Which correlation coefficient is shown in Figures 2 and 3? Please mention it in the caption. I recommend the pearson's correlation coefficient. If the authors prefer spearman, please provide the figures with pearson's correlation coefficient in the supplementary.
        Good suggestion. We will be adding the Pearson correlation maps in the supplement section. The key reason for selecting the Spearman's correlation coefficient is that this correlation is minimally sensitive to outliers and better capture monotonic changes in a 2-variable relationship.

11.     Figure 4: How do the authors interpret negative dlnNd/dlnEXT
        Our explanation is rather simple and guided by the statistically significance of the slopes. Because the negatives slopes are statistically indistinguishable from a zero slope, we treat these negative values as being of a negligible value and no inferences are made about the sign of the slope.

12.      Line 262: dlnLWP/dlnNd is also affected by sampling bias due to missing cloud properties in MODIS as a result of retrieval failure, particularly the positive dlnLWP/dlnNd response (Choudhury and Goren, 2025).

Yes, a sampling bias is certainly a possibility. However, we argue that the quantification of a sampling bias is not possible because assumptions need to be made about the cloud retrievals for those missing pixels. The major issue is validating those assumptions, which, realistically, cannot be done with satellite data only. Satellite simulators and synthetic cloud fields generated by a (cloud) model might help answer this question, but we are not aware of studies that have conducted this type of research. In the revised manuscript, we are going to briefly discuss the potential sampling bias, primarily over regions with the presence of shallow cumulus clouds.

13.      I suggest the authors provide a supplementary figure showing dlnNd/dln(EXTsurface) and dlnNd/dln(AOD)?

The new figures will be provided in the supplement and will be briefly discussed in the manuscript.

14.      A general observation from Figures 4 and 9 is low or negative ACI index over pristine oceans. Can the authors comment on why this could happen in both CALIPSO and MODIS retrievals?

A physical mechanism that could explain the low ACI index over pristine oceans is turbulence in the boundary layer. Because turbulence directly affects the supersaturation at the cloud base, changes in turbulence could explain varying aerosol activation into droplets even for the same aerosol loading (concentration). This explanation is plausible as regions with low ACI index coincide with areas characterized by low cloud coverage and thus, with a reduced cloud top radiative cooling, leading to weaker boundary layer turbulence.

Another factor is associated with the aerosol type over the open ocean. We used the retrieved lidar ratio from CALIPSO-SODA to determine the impact of aerosol typing. Informed by studies based on Raman lidar and high spectral resolution lidar (HSRL, e.g. Burton et al., 2011), clean marine aerosols can be identified with relatively high confidence for samples possessing lidar ratios (LR) < 30 sr. Similarly, pollution and biomass burning aerosol are characterized by LR > 50 sr. Values between 30 sr and 50 sr corresponds to mixture of multiple aerosols, including dust. The relationship for these 3 aerosol types reveals that the ACI metric increases with LR, with values for polluted aerosol exceeding those for clean marine aerosols. If these clean marine aerosols are dominated by the presence of sea salt, then just a few large particles could be contributing to enhanced aerosol extinction coefficient, and thus weakening the relationship between Nd and aerosol extinction coefficient.

[Figure]

Figure R5: Relationship between Nd and aerosol extinction coefficient for 3 different ranges of aerosol lidar ratios (LR). Clean marine aerosols are identified by LR< 30 sr

Reference:

Burton, S. P., Ferrare, R. A., Hostetler, C. A., Hair, J. W., Rogers, R. R., Obland, M. D., Butler, C. F., Cook, A. L., Harper, D. B., and Froyd, K. D.: Aerosol classification using airborne High Spectral Resolution Lidar measurements – methodology and examples, Atmos. Meas. Tech., 5, 73–98, https://doi.org/10.5194/amt-5-73-2012, 2012.

---

## Author Comment (AC2)

We appreciate the scientific insight of the reviewer's comments and suggestions. His/her report helped clarify sections of the manuscript and put our analysis in the context of other relevant studies in the field of aerosol-cloud interactions and radiative forcing. His/her specific comments are addressed below (highlighted in blue).

I feel many important references were not mentioned in this study. Most points discussed in the Introduction have been already well-documented in previous review papers, e.g., https://doi.org/10.5194/acp-20-15079-2020 and more
recent https://doi.org/10.1029/2022RG000799 and the references therein. Would be nice to acknowledge previous work.

We thank the reviewer for suggesting these relevant studies. Both review articles, Rosenfeld et al. and Quaas et al., are now properly cited in the introduction. Quaas et al. is of particular interest, as the article nicely summarizes the challenges in the quantification of ACI and the importance of applying observational constraints to improve current estimates of radiative forcing.

For the effect of retrieval bias, I don't think (Varnai and Marshak, 2009) really touched the effect on ACI, instead more about AOD error. A detailed investigation can refer to https://doi.org/10.5194/acp-22-7353-2022, which may be more relevant here. Regarding the aerosol retrieval issue in low aerosol conditions, a reference should be provided (see the discussion about this in above two review papers).

Correct, Varnai and Marshak did not specifically address the effect of aerosol biases on ACI. We appreciate the reference suggested by Referee 2. In the revised version, we will be adding the following sentence. "Analysis of passive satellite aerosol and cloud retrievals reveal that biases in AOD can yield underestimations of the $N_d$-AOD regression of at least 3% (Jia et al., 2022)".

We agree with the reviewer about limitations for regions with low aerosol loading. We will provide additional discussion in section 4.

Figure. 2: Similar plots but showing sample size would be help here. Also, it's interesting to see negative signals in some regions, particularly in (b) this seems to be more visible in regions with strong precipitation. Any explanation about this? As I can understand these plots were for all clouds; would be interesting to look at non-precipitating clouds only. Similarly, for L233-237: The easiest way to investigate the effect of precipitation on ACI is making the similar plots as Fig. 4 but distinguishing non-precipitating and precipitating clouds

Samples size figures are now included in the manuscript (Fig. R1). In addition, we follow the reviewer's suggestion of separating the analysis into precipitating and non-precipitating samples (also recommended by Reviewer1). To that end, we used the probability of precipitation (POP) defined as the fraction of precipitation CloudSat pixels (reflectivity>-15dBZ) relative to the total cloudy pixels along the 25 km segment. Because precipitation is somewhat patchy in boundary layer clouds, we select a POP>0.3 for defining precipitating scenes. Conversely, non-precipitating segments are defined as having POP <0.05 (Fig. R2). Generally speaking, ACI for precipitating samples decreases, primarily encompassing regions over the open ocean in the subtropics and midlatitudes. We also note that ACI substantially decrease south of Australia, where

ACI is statistically zero. In contrast the same coastal region exhibits ACI up to 0.25 for non-precipitating samples.

[Figure]

Figure R1: Number of 25-km segments used for the derivation of the ACI index.

[Figure]

Figure R2: ACI index map. Black dots indicate grids that are undistinguishable from zero according to a student's t test at 95% confidence level. a) Grids with probability of precipitation (POP) > 0.3, b) Grids with POP<0.05.

L213: maybe an explanation on Spearman correlation would help. It's confusing that the data in Fig. 3a apparently are not concentrated around the regression line, but r_s are mostly larger than 0.95 and even being 1. Please clarify.

The Spearman correlation is less affected by outliers and primarily captures the monotonic increase of the relationship. In the revised manuscript, we will report both Spearman and Pearson correlations.

L216-219 (also the argument on 'S-shape' in abstract): I'm not sure how much I can be convinced by this statement.
- I feel the reason why we didn't see a clear 'flat curve' in high $\sigma$ is the insufficient samples there; binning data into same sample-size bins induces a weak representativity where data are sparse. Even with sparse data, we still see the saturated Nd when $\sigma$ starts going beyond 0.2-0.3. Thus, the analysis here is not sufficient to demonstrate the S-shape is non-physical.
-Even using boundary-layer SO4 (closer to $\sigma$ here), the sigmoid shape is still quite clear (Fig. 1b in https://doi.org/10.5194/acp-23-4115-2023). A recent study further provided the observational evidence for this sigmoid curve based on long-term trends (https://doi.org/10.1038/s41558-023-01775-5). These should be discussed.

- I'd suggest formulating it in a way that the non-lineaer behavior reported in earlier studies tends to be less pronounce when using cloud-base extinction than column AOD, instead of saying it's non-physical as the results presented cannot justify this strong statement.

The way the binning is conducted could certainly change the shape of the curve. We did try multiple bin sizes and the results remained unchanged. With the available dataset, the binning in n-tiles is the appropriate approach to faithfully represent the sampling distribution and reduce the effect of outliers. As suggested by the reviewer, it is pertinent to cite Jia and Quaas (2023) in the discussion section, however, we do not have a hypothesis for why S-shape appears to be present when applied to sulfate. In general, relationships between Nd and other aerosol proxies do not necessarily capture the same physical information, processes, or biases; and, therefore, our conclusions are only valid for optical aerosol properties derived from satellites. However, the contrast between the sigmoid curve for AOD and a semi-linear relationship for aerosol extinction is quite evident, and the saturation of the curve for high AOD is not observed for aerosol extinction coefficient. However, we will revise the text to convey the idea that the non-linearity is less pronounced when vertically-resolved aerosol extinction coefficient is used in the analysis.

L260-261: This is an interesting point. The authors could demonstrate this even clearer by making a panel (b) for Fig 7 but lumping all global data together.

The updated figure is included below, with red diamond representing global data, and will be discussed in the revised version.

[Figure]

**Minor comment:**
Since only the vertical co-location of cloud and aerosol layers is studied, the term 'Progress' in the title seems too broad and gives the impression of a review-like paper. I suggest removing it.

Following the reviewer's suggestion, we modify the title to: "Advancing the quantification of aerosol-cloud interactions with the use of the CALIPSO-CloudSat-Aqua/MODIS record"

L26: Estimates -> Observational estimates
Corrected, thanks

This work largely follows Painemal et al. (2020). The importance of using vertically collocated aerosol has been already well-justified. Would be good to explain what new message one could get beyond the existing literature.

Unlike Painemal et al. (2020), the global quantification of ACI and susceptibilities is a key contribution of our manuscript. Also, the integrated study of cloud susceptibilities and ACI using MODIS-CALIOP-and CloudSat is also another contribution. We will be highlighting these points in the revised manuscript.

L68: 'shortcomings': Since the text so far only highlights the benefits, it might be helpful to flag what shortcomings the readers can expect next.

Correct, we are going to discuss some outstanding issues, primarily associated with the fact that optical properties differ from aerosol concentration.

' metrics of cloud susceptibilities of ACI' and 'cloud susceptibility' are the same, aren't they?
Correct, we modified the sentence to read: "metrics of ACI and cloud susceptibility"

L80-82: this sentence is hard to read. Please explain what 'This choice of CALIPSO-based dataset responds to limitations of the standard CALIPSO product' means

In short, our research version of the CALIPSO retrievals are estimated by solving the lidar equation, using an iterative process that finds both the extinction coefficient and lidar ratio that matches an independent AOD, derived from the SODA algorithm. This eliminates the need of classifying aerosols into specific aerosol types and assuming a constant lidar ratio. Because the lidar ratio is highly variable, assuming a constant lidar ratio introduces significant uncertainties in the CALIPSO aerosol product. Painemal et al. (2019) show that that this new aerosol extinction coefficient better compares with airborne observations from a high spectral resolution lidar. In the revised version, we are going to explicitly explain the advantages of our dataset over the standard CALIPSO products.
Reference:
Painemal, D., Clayton, M., Ferrare, R., Burton, S., Josset, D., and Vaughan, M.: Novel aerosol extinction coefficients and lidar ratios over the ocean from CALIPSO–CloudSat: evaluation and global statistics, Atmos. Meas. Tech., 12, 2201–2217, https://doi.org/10.5194/amt-12-2201-2019, 2019.

L93: I personally think CTH is a better term than ZT for cloud top height, which has been widely used. Would be easier for readers

We generally use a simple notation instead acronymous because the variables are easier to express and understand in equations.

L99: 'height': is it cloud top height?
Correct, all the variables in the sentence are for clouds.

Eq 1: Though the authors referred to (Albertcht et., 1990), it's good to provide the full formulation here along with all parameter values need in the calculation so that people can easily follow.

The formulation is a bit more complicated than a single formula solely depending on temperature and pressure. Because the expression has been utilized in a number of studies, we refer the reader to Albrecht et al.

112-113: How it can categorize the low-cloud precipitation rate is not clear. I guess the authors put a 'raining' flag if Zmax>-15, otherwise 'non-raining', right?

We appreciate the reviewer's comment. Yes, we identify precipitation at the CLoudSat pixel level as Zmax>-15dBZ. The categorization is not discussed in the later sections, thus we removed the sentence "The impact of additional precipitation categorization (drizzle: -15< Zmax<=-7), light rain: -7< Zmax<=0, and rain: Zmax>0) is discussed in Section 4."

L124: Could you clarify what you mean by "the closest CloudSat CPR pixels to the 25-km line"? What exactly does the 25 km line refer to here? L134-136: it's a bit unclear if the cloud-top height is from MODIS or CALIOP as stated earlier? To match and 1x1 modis pixels I'd assume it's from MODIS right?

It refers to the closest pixels (and ground-track) to the CALIPSO ground-track, represented by the 25 km segment in Figure 1 (in blue). The derivation of cloud layer height for computing aerosol extinction coefficient is from CALIOP. Filtering of MODIS pixels are based on MODIS cloud top height. We are going to clarify this in the revised manuscript.

L140: the threshold is generally set to 4; could you explain why 2 is used here? Does it mean more optically thin clouds are included in this study?

The rationale is based on the Painemal et al. (2025). Using airborne polarimetric retrievals, Painemal et al (2025) show that retrievals tend to be more robust for optical depth greater than 2, especially if the satellite data correspond to a cloudy scene (high cloud coverage).

Reference:
Painemal, D., Smith, W. L. Jr., Gupta, S., Moore, R., Cairns, B., McFarquhar, G. M., & O'Brien, J. (2025). Can we rely on satellite visible/infrared microphysical retrievals of boundary layer clouds in partially cloudy scenes? implications for climate research. *Geophysical Research Letters*, 52, e2024GL113825. **https://doi.org/10.1029/2024GL113825**

Eq3: Only data with CF>80% are analyzed. In this case, S_CF cannot reflect the real effect. would be nice to mention this limitation here though it appears quite later in the paper

Correct. We will emphasize the fact that the changes in cloud coverage due to aerosols cannot be investigated because the different filters applied in the analysis, by design, remove a substantial part of the aerosol-cloud fraction relationship.

L207: 'is constrained using AOD' what does this mean? Is $\sigma$ vertically integrated into the value of AOD?

Correct, the vertically integrated extinction coefficient is AOD. We modified the sentence to read: It is noteworthy to mention that because the vertically integrated $\sigma_{ext}$ in the CALIOP-S data product is AOD…"

Fig.7: It would be easier to follow if the authors marked these 4 regions in Fig 6.
Good suggestions. This will be implemented in the revised version.

l261-262: I think the story in Arola et al. (2022) is quite different to the argument here. They attributed the invert-V to retrieval errors. Citing this paper here seems a bit confusing unless the authors make this clear.

The reviewer is correct in the sense that Arola et al. (2022) address a number of potential biases, including retrieval uncertainties. However, the article also explores the effect of cloud natural heterogeneity and discusses how spatial changes can yield a relationship that are not necessarily the manifestation of the aerosol indirect effect. This concept is partially encapsulated in the article title ("Aerosol effects on clouds are concealed by natural cloud heterogeneity and satellite retrieval errors").

L266: It's Intuitive that the product of S_nd_lwp (Iig. 6a) and ACI (Fig. 4a) should be negative as their signs are opposite, especially in Tropics; so it's kinda surprising that it turns to be negligible. Could the authors explain this a bit more?

From a point of view of the statistical estimation, this is the result of having a negligible ACI index for regions with positive susceptibility of LWP to Nd. The physical interpretation of these results is, nevertheless, challenging. Cognizant that these estimates need to be validated with other datasets and methods, we interpret this negligible susceptibility as the modest effect of aerosols to modify the relationship between Nd and LWP for those specific regions. This will be discussed further in the article, including potential sources of uncertainties that can also challenge our interpretation.

L298: I'd avoid words like "novel" or "new," or anything implying the study is the first to show a particular conclusion. A more neutral phrasing would work better. It would be easier for readers to follow the results if Fig. 4 were placed as a separate panel within Fig. 9; so that readers would not need to scroll back and forth. And why does the ACI(based on AOD) index appear to be negative here? It's overall positive in previous studies. Would be good to discuss.

We agree on the use of "novel" or "new" and the inclusion of Fig.4 as a subpanel of Fig. 9. We note that regions with seemingly negative slopes are statistically indistinguishable from zero and, therefore, the slope sign is not discussed in the article.

L301-303: The authors stressed a lot on the difference in ACI between $\sigma$ and AOD; but it's very important to mention here that in the end we care about anthropogenic perturbation of Nd and forcing which also relies on PI-PD change in the utilized proxy, not the simple slope (https://doi.org/10.1029/2022RG000799).

This is a fair criticism. While assessing PD-PI changes is important for understanding the anthropogenic forcing, these satellite-based metrics are also useful for evaluating climate models (e.g. Zheng et al., 2025). Also, anthropogenic forcing estimates benefit from quantifying these slopes (e.g. Bellouin et al., 2021).

Reference:

Zheng, X., Feng, Y., Painemal, D., Zhang, M., Xie, S., Li, Z., Jacob, R., and Lusch, B.: Regime-based aerosol–cloud interactions from CALIPSO-MODIS and the Energy Exascale Earth System

Model version 2 (E3SMv2) over the Eastern North Atlantic, Atmos. Chem. Phys., 25, 17473–17499, https://doi.org/10.5194/acp-25-17473-2025, 2025.

L311-312: the use of AOD doesn't misguide the modelers as long as they are looking at AOD as well. I think this sentence can be dropped.

It could still misguide modelers in the sense that the relationship between AOD and Nd could point to processes and covariations that are not necessarily related to aerosol-cloud interactions. We will slightly modify the sentence to reflect our response.

L311: I find the phrase "unlike previous assessments, but similar to Gryspeerdt et al. (2016)" a bit confusing. Gryspeerdt et al. (2016) is also a 'previous' study, so it might help to clarify what you mean here. For example, do you refer to a specific group of 'previous assessments' using a different methodology?

We agree with the reviewer. Because we are talking about methodological differences rather than differences in conclusions, we are going to omit the "unlike previous assessments" phrase.